# Execution-Based Evaluation for Open-Domain Code Generation

**Zhiruo Wang   Shuyan Zhou   Daniel Fried   Graham Neubig**
Language Technologies Institute, Carnegie Mellon University
{zhiruow,shuyanzh,dfried,gneubig}@cs.cmu.edu

## Abstract

To extend the scope of coding queries to more realistic settings, we propose ODEX, the *first* Open-Domain EXecution-based natural language (NL) to Python code generation dataset. ODEX has 945 NL-Code pairs spanning *79 diverse libraries*, along with *1,707 human-written test cases* for execution. Our NL-Code pairs are harvested from StackOverflow forums to encourage *natural* and *practical* coding queries. Moreover, ODEX supports *four* natural languages as intents, in English, Spanish, Japanese, and Russian. ODEX unveils intriguing behavioral differences among top-performing code language models (LM). While CODEX achieves better overall results, CODEGEN improves effectively via scaling – CODEGEN 6.1B performs comparably with CODEX 12B. Both models show substantial gaps between open and closed domains, but CODEGEN gaps tend to decrease with model size while CODEX gaps increase. We release ODEX to facilitate research into open-domain problems for the code generation community.[1]

## 1 Introduction

Evaluations of NL-to-code generation systems, especially for general-purpose programming languages such as Python, have put an increasing emphasis on methods that execute code to verify the results. The predominant approach for creating such test sets is to manually write test cases for canonical code solutions (Chen et al., 2021; Austin et al., 2021; Lai et al., 2022; Huang et al., 2022). The correctness of model predictions is then evaluated by seeing if generated code passes the test cases (Chen et al., 2021). Compared to execution-free metrics such as text match against reference solutions, execution-based methods more rigorously assess the functional correctness of code (Hendrycks et al., 2021; Chen et al., 2021).

However, most resources with execution support only apply to *closed-domain* code, that only use Python built-in functions (Chen et al., 2021; Hendrycks et al., 2021; Austin et al., 2021; Li et al., 2022; Haluptzok et al., 2023) or specific libraries in data science domains (Lai et al., 2022; Huang et al., 2022). This focus on closed-domain problems diverges substantially from natural *open-domain* program usage covering *a diverse range of libraries and functionalities* (Yin et al., 2018; Agashe et al., 2019; Wang et al., 2023). To enable execution-based evaluation for coding queries using libraries, we present ODEX, an Open-Domain EXecution-based dataset (§2). We build ODEX by creating 1,707 test cases for 945 NL-Code pairs from the CoNaLa (Yin et al., 2018) and MCoNaLa (Wang et al., 2023) datasets, both stemming from Stack-Overflow[2] with broad practical coding queries.

We analyze and highlight three aspects of ODEX (§3). First, ODEX has broad domain coverage of 79 libraries, with $53.4\%$ of the problems employing at least one library. Second, ODEX contains queries in four different languages, with 439, 90, 164, and 252 samples in English, Spanish, Japanese, and Russian, as shown in Figure 1. Third, ODEX addresses three unique challenges in open-domain code execution: irreproducible runs (Figure 1 a), randomized outputs (Figure 1 b), and specialized equivalence checks (Figure 2).

We evaluate two state-of-the-art code LLM families, CODEX and CODEGEN, on ODEX (§5). Our study shows that larger model sizes and augmented training data improve execution accuracy. Meanwhile, we observe satisfactory multilingual capabilities, despite that neither model was specifically designed for multilingual usage. However, we find that models face greater yet varied challenges with open-domain queries compared to closed-domain queries (§5). Specifically, CODEX achieves higher

---

[1] https://anonymous.4open.science/r/odex-emnlp

[2] https://stackoverflow.com

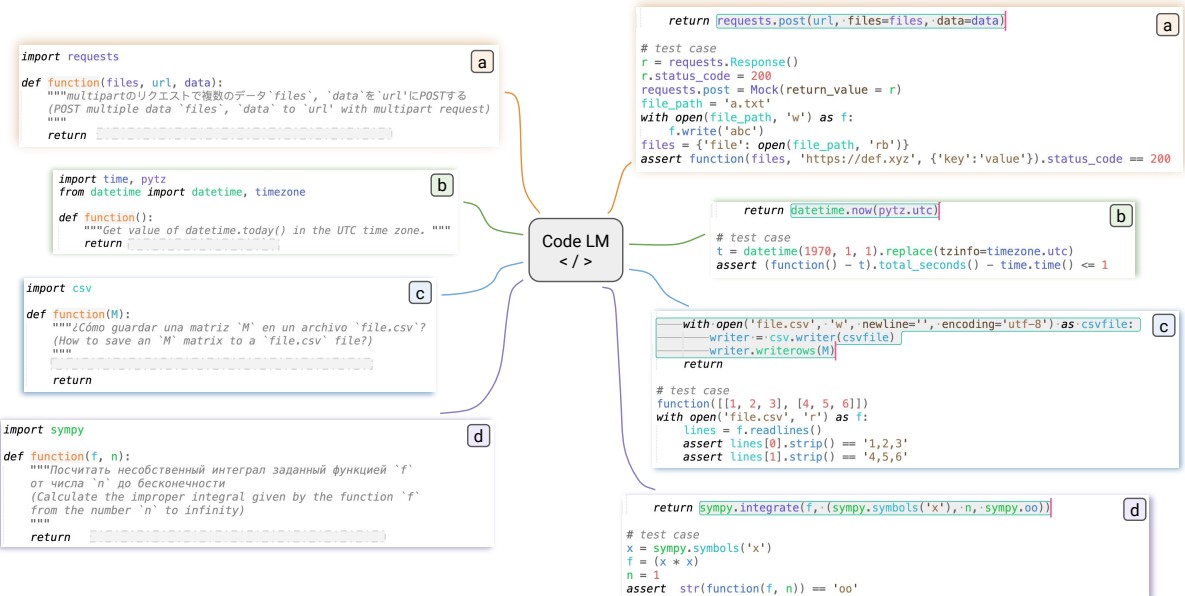

Figure 1: Examples in the ODEX dataset. Inputs on the left are function-formatted with (1) library import expressions; (2) function signatures that declares the function name and input arguments; and (3) natural language intents as part of the docstrings (English translations are not included in the actual non-English inputs during inference). Gray boxes indicate places for code solutions. As shown on the right, a code LM fills out the gray boxes with code solutions, which are then executed on the unit tests underneath. Notably, writing unit tests for open-domain queries is often more challenging: a requires simulated execution due to the difficulty of reproduction; b is verified through approximate equivalence. Prior work focuses more on basic assertions, as in c and d .

overall results, while CODEGEN presents better parameter efficiency and more balanced open-closed domain performance as model size scales up. By comparing execution-based metric with a series of execution-free metrics (§6), we further confirm the advantage of execution on allowing alternative solutions, but also show the potential of lexical metrics to identify simple bug fixes.

ODEX jointly facilitates *practical open-domain* code generation and *execution-based* evaluation. It serves as a comprehensive data benchmark for NL-to-code systems, supporting diverse NL contexts, library usage, and evaluation methods. By addressing the unique challenges of test creation and execution, we hope to lay a foundation for evaluating open-domain code via execution.

## 2 The ODEX Dataset

In this section, we describe our four-step process of constructing the ODEX dataset. We first collect resources of natural, open-domain coding queries (§2.1). Next, we establish the annotation standard and procedures for test case creation (§2.2). We then describe the annotator hiring and working processes (§2.3). Finally, we conduct checks to ensure data quality (§2.4).

### 2.1 Resource Collection

We take two NL-to-code datasets, CoNaLa (Yin et al., 2018) and MCoNaLa (Wang et al., 2023), as sources for ODEX. We refer to them together as (M)CoNaLa. Their NL-Code pairs are collected from StackOverflow, which contains abundant coding queries that (1) naturally reflect practical program usage, and (2) cover diverse domains as measured by libraries used. These properties align well with our main focus on open-domain queries. (M)CoNaLa further proofs and clarifies its NL intents using human annotators to ensure data quality.

### 2.2 Annotation Standard and Procedures

Given each source NL-Code pair, our main annotation task is to write test cases to check code execution correctness, as illustrated by the four steps in Figure 2. A qualified test case should verify the main functionality of the canonical code solution. In the case where annotators do not understand the language of the intent, we use translation tools such as the Google Translate API.[3]

***Step 1*: Wrapping Snippets into Functions**
Code solutions in (M)CoNaLa are often short snippets (e.g., x = np.zeros(5)) to ensure more pre-

---
[3]https://translate.google.com

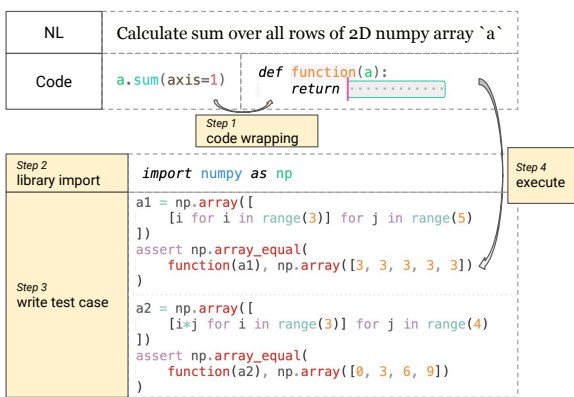

| NL | Calculate sum over all rows of 2D numpy array `a` |
| Code | `a.sum(axis=1)` `def function(a): return ............` |

```
                          Step 1
                          code wrapping

Step 2
library import    import numpy as np

                  a1 = np.array([
                      [i for i in range(3) for j in range(5)
                  ])
                  assert np.array_equal(
                      function(a1), np.array([3, 3, 3, 3, 3])
                  )
Step 3
write test case   a2 = np.array([
                      [i*j for i in range(3) for j in range(4)
                  ])
                  assert np.array_equal(
                      function(a2), np.array([0, 3, 6, 9])
                  )

Step 4
execute
```

Figure 2: An example annotation comprising four steps.

cise matches with NL intents, but to be executable they often need additional context such as variable assignments. We therefore wrap code into standalone functions by specifying input and output arguments as contexts. For example, *Step 1* in Figure 2 identifies variable a as an input argument.

***Step 2*: Specifying Library Prerequisites**  Due to the open-domain coverage of (M)CoNaLa, some code snippets require extra library imports to execute correctly. Accordingly, our second step is to specify the prerequisite libraries for code solutions.

***Step 3*: Test Case Annotation**  Next, we write test cases that contain three parts: (1) input: passing values to input arguments, (2) output: stating expected execution outputs, and (3) assertion: checking if execution results match the expected outputs.

However, test case creation for open-domain code faces three challenges. First, safe and reproducible execution can be hard to achieve. As in Figure 1 a , it is impractical to send an HTTP request when evaluating this sample. Instead, we use mock to simulate the output (a success response status code 200). Second, some codes entail randomness (e.g., random.randint(3,5)) and have no definite value. We instead make bounding assertions, e.g., checking that all elements are integers within the range of [3,5]. Third, standard equivalence checks by == may be invalid, since library-specific objects often require specialized equality checks. For example, checking the equivalence of two NumPy arrays a and b uses np.array_equal(a,b), while a == b would cause execution errors.

***Step 4*: Self Verification**  In the last step, we perform self-verification to efficiently ensure the annotation quality. We execute the canonical code solution on each newly created test case. Unless the

test case enables a successful pass of the solution, it should not be taken as a valid annotation.

## 2.3 Annotator Hiring and Task Fulfillment

As our data involves diverse functionalities from multiple libraries, our annotation task holds a relatively high standard for annotators. A qualified annotator should be proficient in Python and common libraries, and in writing workable test cases.

We chose to hire undergraduate students who have strong computer science backgrounds in Python. Of the 20 applicants who applied, we first conducted a resume screening to filter candidates with sufficient programming experience. Next, we gave each candidate an annotation test with five randomly selected NL-Code pairs. Since the test mirrors the official annotation process, we provided clear instructions about each step (as in §2.2) and code scripts for self-verification. Candidates were asked to finish their tests in three calendar days. Based on their test performance, we hired four candidates to officially participate in this job.

## 2.4 Quality Check

We put great effort into ensuring data quality throughout the annotation process. To assist annotators in more efficiently and accurately writing workable test cases, we require them to execute each written test case using the verification code that we provided, and explicitly report whether the canonical code solution can successfully pass all the annotated test cases that they created.

After the annotation, the authors performed post-hoc verification to check if each test case reads reasonably and executes correctly. In our final rounds of automatic quality checks, we confirm that the pass rate for all canonical code solutions over their annotated test cases is 100%.

We collect a total of 945 samples with NLs in four languages, including 439 samples in English, 90 in Spanish, 164 in Japanese, and 252 in Russian.

## 3 Dataset Analysis

We analyze ODEX from three aspects: domain diversity (§3.1), sample complexity (§3.2), and execution support (§3.3).

## 3.1 Diversity

One unique property of ODEX is its broad domain coverage. We categorize codes that entail library usage (both built-in and third-party) as being in the

*open domain* and those with none in the *closed domain*. Different libraries often serve specific functions and have unique capabilities. For instance, the `datetime` library is designed to handle date/time operations, while other libraries focus on various other fields such as data analysis or web requests. Therefore, in this work, we view the diversity in libraries as a representation of distinct domains.

| Language | # Unique Libraries | Size | | |
|---|---|---|---|---|
| | | Open | Closed | Total |
| en | 45 | 230 | 209 | 439 |
| es | 20 | 48 | 42 | 90 |
| ja | 44 | 113 | 51 | 164 |
| ru | 35 | 114 | 138 | 252 |
| Total | 79 | 505 | 440 | 945 |

Table 1: Number of open- and closed-domain examples, and number of libraries involved in each language.

Table 1 reports domain statistics and Figure 3 shows the library distribution. ODEX covers a diverse set of 79 libraries, which varies per language. Most samples, 53.4%, use at least one library.

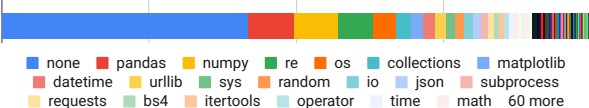

Figure 3: ODEX library distribution.

**Comparison to Existing Datasets** We compare ODEX with eight other code generation datasets that support test case execution: HumanEval (Chen et al., 2021), MBPP (Austin et al., 2021), APPS (Hendrycks et al., 2021), MTPB (Nijkamp et al., 2023), P3 (Haluptzok et al., 2023), DSP (Chandel et al., 2022), DS-1000 (Lai et al., 2022), and Exe-DS (Huang et al., 2022).

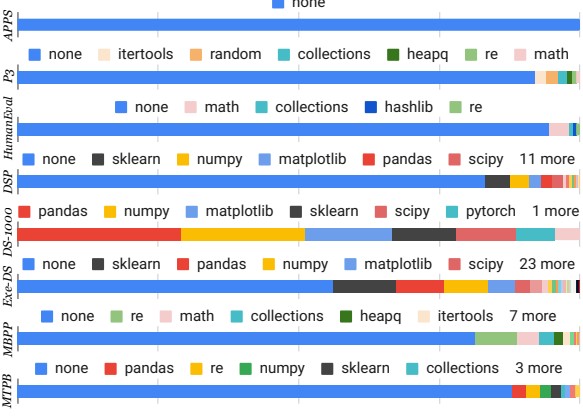

Figure 4: Library distribution of eight other datasets.

From their distributions in Figure 4, six out of eight datasets focus on the closed domain and most

examples use zero libraries. Such examples deviate from realistic programs, which often use APIs of different libraries. DS-1000 and Exe-DS feature some open-domain problems, but their library usage is more homogeneous with a particular focus on data science domains. Moreover, DS-1000 restricts to code using libraries but only has seven libraries. In contrast, ODEX is more "colorful"; it covers significantly more open-domain libraries, as well as frequent queries in the closed domain.

**Comparison to Natural Distribution** To provide a reference on natural domain distribution, we approximate real-world usage by counting GitHub Python files that use each library. As shown in Figure 5, ODEX presents a better alignment with the practical scenario concerning the open domains – it features more diverse domains and preserves the long-tailed pattern in practical scenarios.

The full lists of libraries and their frequencies about ODEX, the eight comparison datasets, and the approximated natural setting are in §A.1.

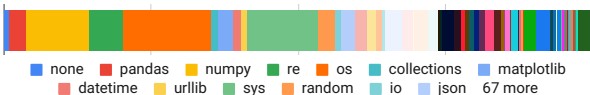

Figure 5: Approximated natural distribution based on GitHub Python files in the open domain.

## 3.2 Complexity

To measure dataset complexity, we first calculate the lengths of NL intents and code snippets. We tokenize NL intents with the spaCy[4] tokenizers in respective languages; we follow Yin and Neubig (2018) to tokenize code. For code, we also parse the AST tree using the Python standard `ast` library,[5] and count the number of input and output variables to quantify the complexity of execution contexts.

| Language | len(NL) | len(Code) | depth(AST) | $N_{in}^{var}$ | $N_{out}^{var}$ |
|---|---|---|---|---|---|
| en | 14.36 | 18.49 | 7.02 | 1.13 | 0.21 |
| es | 18.69 | 28.62 | 7.74 | 1.46 | 0.64 |
| ja | 17.24 | 17.70 | 6.77 | 1.40 | 0.41 |
| ru | 11.39 | 20.19 | 6.94 | 1.44 | 0.71 |

Table 2: Complexity measured in the averaged number of NL words, code tokens, AST depth, and i/o variables.

In Table 2, we see that code in the Spanish set is longer on average than other languages. For both the input and output sides, code in the English set has fewer variables, suggesting potentially simpler

---

[4]https://spacy.io/
[5]https://docs.python.org/3/library/ast.html

| Dataset | Samples | Domain | Executable? | Avg. Test Cases | Data Source | NL |
|---|---|---|---|---|---|---|
| JuICe (Agashe et al., 2019) | 1,981 | open | ✗ | - | GitHub Notebooks | en |
| HumanEval (Chen et al., 2021) | 164 | 4 | ✓ | 7.7 | Hand-written | en |
| MBPP (Austin et al., 2021) | 974 | 8 | ✓ | 3.0 | Hand-written | en |
| APPS (Hendrycks et al., 2021) | 10,000 | 0 | ✓ | 13.2 | Competitions | en |
| DSP (Chandel et al., 2022) | 1,119 | 16 | ✓ | 2.1 | Github Notebooks | en |
| MTPB (Nijkamp et al., 2023) | 115 | 8 | ✓ | 5.0 | Hand-written | en |
| Exe-DS (Huang et al., 2022) | 534 | 28 | ✓ | - | GitHub Notebooks | en |
| DS-1000 (Lai et al., 2022) | 1,000 | 7 | ✓ | 1.6 | StackOverflow | en |
| CoNaLa (Yin et al., 2018) | 2,879 | open | ✗ | - | StackOverflow | en |
| MCoNaLa (Wang et al., 2023) | 896 | open | ✗ | - | StackOverflow | es, ja, ru |
| ODEX | 945 | 79 | ✓ | 1.8 | StackOverflow Hand-Written | en, es, ja, ru |

Table 3: Comparing ODEX with other NL-to-code generation datasets, in terms of domain diversity (*Domain*), test-case execution support (*Evaluation*, *Avg. Test Cases*), and natural language contexts (*NL*). Since it is hard to calculate the exact number of libraries for some open-domain datasets that do not specifically import required libraries in the code, we mark their domains as *open* instead of providing the exact number of domains.

execution environments, which could stem from relative simplicity of SO queries asked in English.

### 3.3 Execution Support

We systematically compare code generation datasets that concern execution or open-domain code in Table 3. ODEX is the first dataset that supports execution-based evaluation for open-domain code. While ODEX does not have the largest number of test cases, we discuss in §7 how these test cases can still reliably measure code correctness.

## 4 Experiment Setup

Code LLMs have achieved strong results on multiple code generation tasks, yet their open-domain proficiency is understudied due to the limited domain settings of past datasets. To examine model capabilities in the open domain, we evaluate two top-performing model families, CODEX and CODE-GEN, on ODEX. We perform evaluations using a prompting setting, without finetuning any model.

We introduce the baseline models, the prompt settings, and lay out the metrics for evaluation.

**The CODEX Family** At the time of this work, CODEX had three publicly available models. CODE-CUSHMAN-001 (C1) is a 12B CODEX model in Chen et al. (2021). CODE-DAVINCI-001/002 (D1, D2) are two 175B GPT-3 models.[6]

**The CODEGEN Family** CODEGEN (Nijkamp et al., 2023) models are auto-regressive models trained on a combination of NL and code corpora, differing in model sizes (350M, 2.7B, 6.1B, 16.1B) and training data. Models are progressively trained

[6]https://beta.openai.com/docs/model-index-for-researchers

on THEPILE (Gao et al., 2020), BIGQUERY,[7] and BIGPYTHON datasets are denoted as NL, MULTI, and MONO. The most powerful CODEGEN-16.1B-MONO, performs similarly to CODE-CUSHMAN-001 on the HumanEval and MTPB datasets.

**Prompt Design** For fair comparison, we use the same prompt for both model families. While prompting with few-shot in-context examples may improve, our experiments do not always find this helpful for both models. Therefore, we report *zero-shot* results as baselines and leave few-shot results to §7. Creating zero-shot prompts only requires content from the test sample. Following Chen et al. (2021), we construct prompts by concatenating function context and a docstring. A docstring includes the NL intent and optional unit tests (compared in §7). Figure 6 shows an example prompt.

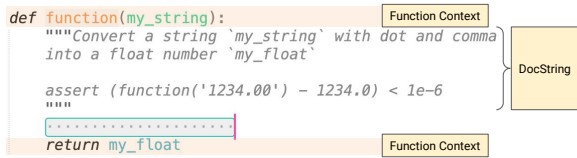

Figure 6: Zero-shot prompt with one test case in docstring. The gray box notes the place for code solution.

**Evaluation Metrics** We follow Chen et al. (2021) and measure the execution accuracy using the *pass@k* metric, by computing the fraction of problems having at least one correct prediction within $k$ samples. We also compare it with a series of execution-free metrics later in §5.

**Implementation Details** We follow Chen et al. (2021) and use nucleus sampling (Holtzman et al., 2020) with top-$p$ set to 0.95 and temperature set to 0.8. We set outputs to a maximum of 512 tokens.

[7]https://cloud.google.com/bigquery

| Language | CODEX | pass@k | | | | CODEGEN | pass@k | | | |
|---|---|---|---|---|---|---|---|---|---|---|
| | | 1 | 2 | 5 | 10 | | 1 | 2 | 5 | 10 |
| en | | 31.91 | 44.67 | 59.95 | 68.79 | | 26.26 | 32.18 | 39.10 | 42.82 |
| es | CUSHMAN-001 | 31.89 | 43.33 | 55.72 | 63.33 | 350M | 16.67 | 21.85 | 27.82 | 30.00 |
| ja | | 25.67 | 36.69 | 49.27 | 57.32 | | 17.44 | 22.86 | 28.21 | 30.49 |
| ru | | 40.00 | 53.48 | 66.63 | 73.41 | | 25.87 | 31.44 | 37.44 | 40.87 |
| en | | 33.62 | 46.65 | 60.18 | 67.43 | | 35.24 | 42.87 | 50.68 | 53.99 |
| es | DAVINCI-001 | 36.89 | 49.46 | 61.37 | 68.89 | 2.7B | 26.00 | 33.65 | 41.52 | 45.56 |
| ja | | 31.04 | 42.11 | 54.26 | 61.59 | | 24.27 | 32.10 | 41.13 | 45.12 |
| ru | | 43.21 | 57.53 | 70.03 | 76.59 | | 39.64 | 48.11 | 57.23 | 61.90 |
| en | | 47.15 | 57.61 | 67.87 | 73.12 | | 34.49 | 37.91 | 41.18 | 43.05 |
| es | DAVINCI-002 | 47.44 | 57.90 | 66.33 | 71.11 | 6.1B | 28.56 | 32.05 | 35.86 | 37.78 |
| ja | | 41.46 | 50.42 | 59.47 | 64.02 | | 35.55 | 40.11 | 44.12 | 46.34 |
| ru | | 51.87 | 63.36 | 73.03 | 78.17 | | 44.64 | 47.29 | 49.82 | 51.19 |

Table 4: Execution accuracy of CODEX and CODEGEN-MONO models.

# 5 Experiment Results

We first present the overall performance of two model families on ODEX (§5.1). Next, given the unique challenges of open-domain code, we study the variances between open- and closed-domain problems (§5.2), and in individual domains (§5.3).

## 5.1 Baseline Performance

**CODEX Results** As in Table 4, aligning to existing works and our intuition, larger DAVINCI 175B models outperform the smaller CUSHMAN 12B model, and the 002 version improves over 001. This trend holds for all languages and all sampling sizes. Somewhat surprisingly, all models attain decent results on non-English problems, even though CODEX is not designed for multilingual use. This high accuracy on non-English problems suggests the multilingual potential of CODEX models.

**CODEGEN Results** We report results of MONO models in Table 4 given their superior performance over NL and MULTI variants (Nijkamp et al., 2023). The pass rate increases as CODEGEN grows from 350M to 2.7B, and continues to increase in non-English languages when further scaling to 6.1B. CODEGEN exhibits multilingual capacity, as its results on non-English subsets are close to that on English, and consistently increase during scaling.

Although CODEX and CODEGEN have comparable performance on existing datasets such as HumanEval, ODEX effectively unveils the efficacy of CODEGEN on open-domain coding queries even with many fewer parameters, i.e., CODEGEN 6.1B yields similar pass@1 to the 176B CODEX DAVINCI-001 model, although not necessarily so when $k$ increases. More detailed results (pass@k at $1 \leq k \leq 10$) for both models are in §B.

## 5.2 Open Domain versus Closed Domain

**CODEX Results** Figure 7 (left) shows pass@1 on open-domain and closed-domain. All CODEX models score much lower in open than in closed domain. Such large gaps hold across all languages, ranging from 4.34 in Spanish to 38.57 in Japanese on the best DAVINCI-002 model. Model upgrades (C1 → D1 → D2) do not always reduce the gaps. Gaps slightly shrink in Spanish, but increase in English and Japanese. While D2 performs the best, it also exhibits the most severe gaps. These findings suggest that common practices to improve LLMs may not address the complexities inherent in open-domain coding problems. It is hence imperative that more advanced strategies are employed.

**CODEGEN Results** As shown in Figure 7 (right), CODEGEN also has substantial gaps between open and closed domains, however, smaller than CODEX gaps across all languages, by on average 6.0% points. As model size increases from 2.7B to 6.1B, the gaps reduce by about 6.3 points in English and 1.7 points in Spanish. This is in contrast to CODEX, which when scaling up to DAVINCI-002, these gaps continue to increase by 4.9 points on average, indicating that scaling up CODEGEN more effectively catches up on open-domain performance.

## 5.3 Domain Variance

We now dive deeper into the results within individual domains. We focus on the CODE-DAVINCI-002 model as it has the best performance across all models. In Figure 8, we plot accuracy with respect to the domain frequency, as approximated in §3.1.

Execution accuracy is not low on all open domains. For example, CODE-DAVINCI-002 achieves 50% pass@1 for several common libraries such as random and math. But high domain frequency does not ensure model proficiency. For example,

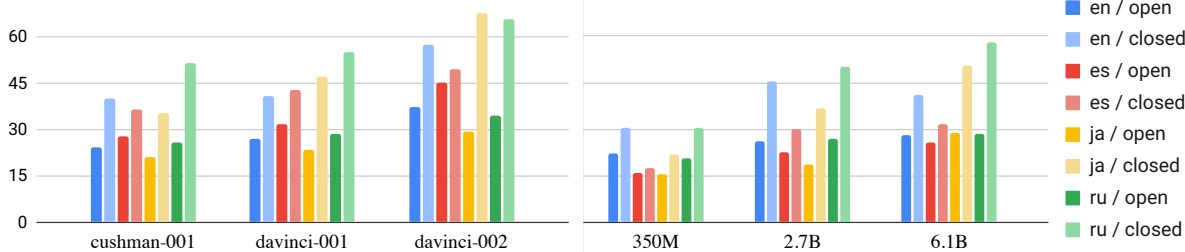

Figure 7: CODEX (left) and CODEGEN (right) *pass@1* on open- and closed-domain problems in each language.

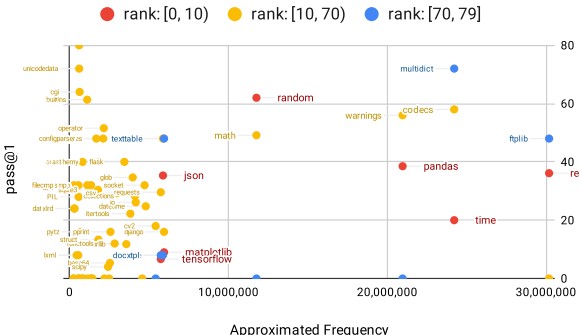

Figure 8: CODEX pass@1 for domains of varied frequencies. Domains are differently colored based on their frequency ranking: the 10 most frequent domains in red, the 10 least frequent domains in blue, and other domains in the middle in yellow.

on libraries with complex functionalities such as `matplotlib` and `tensorflow`, *pass@1* can go below $10\%$. See §C for more domain-wise results.

## 6 Comparing to Execution-Free Metrics

In this section, we study the alignment between execution-based evaluation and five execution-free metrics, identifying advantages for both types.

**Model Ranking Using Different Metrics** We evaluate models using five execution-free metrics using lexical, syntax, and semantic matches: BLEU (Papineni et al., 2002), ROUGE (Lin, 2004), METEOR (Banerjee and Lavie, 2005), ChrF (Popović, 2015), and CodeBLEU (Ren et al., 2020). Refer to §D.1 for more descriptions.

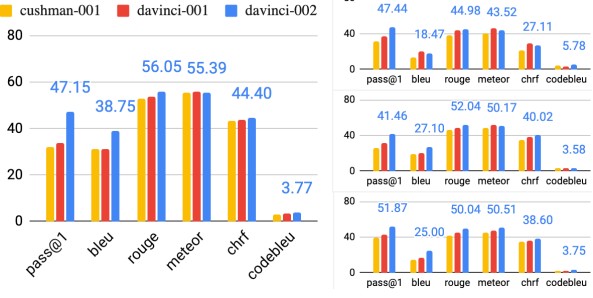

Figure 9: CODEX models evaluated on six metrics.

We analyze using CODEX, given its better per-

formance. As shown in Figure 9, model rankings by execution-free metrics do not precisely correlate with their rankings by execution accuracy. Even when the rankings align, their differences are largely not proportional. Comparing the metrics, ChrF and METEOR have smaller inter-model variances, while BLEU and ROUGE change more and correlate better with pass rates. Notably, Code-BLEU is low in most settings and might not be suitable for evaluating code in snippet-style.

**Metric Correlation** We next evaluate whether execution-free metrics might be used to discriminate between passed and failed samples. We take BLEU as an example since it shows similar ranking patterns to execution. Figure 10 shows negligible variances in BLEU scores of passed and failed groups. The other four metrics exhibit similar patterns, as could be found in §D.3.

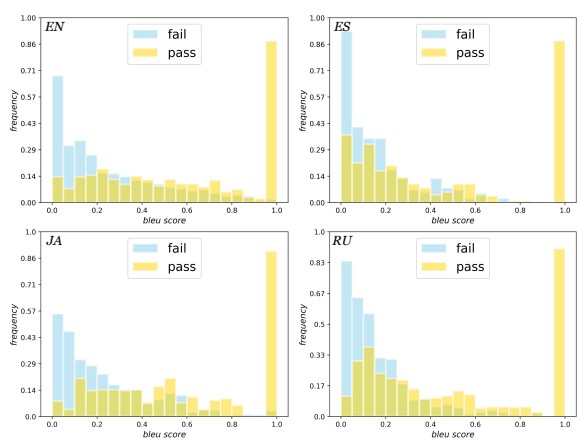

Figure 10: BLEU scores on passed and failed samples.

## 7 What Affects Model Performance?

Besides differences in model configurations, we study three factors that might affect performance.

**Number of In-Context Examples** Models might benefit from example NL-Code pairs. We thus explore to few-shot prefixing $N \in \{1, 2, 3\}$ input-output pairs in prompts. In Figure 11 (left), for CUSHMAN-001 and DAVINCI-001, few-shot ex-

amples yield a clear improvement over the zero-shot setting; but for the strongest DAVINCI-002, it brings minimal gains in English. See similar results in other languages in §E.1.

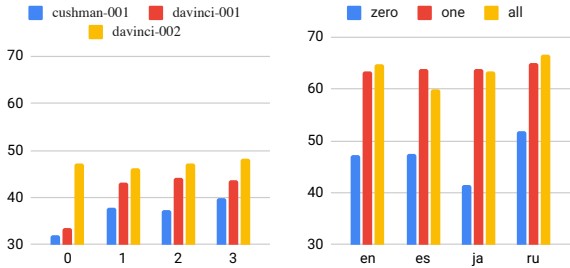

Figure 11: Left: CODEX *pass@1* (on English set) using 0/1/2/3-shot prompts. Right: DAVINCI-002 *pass@1* when adding zero, one, or all test cases in prompts.

**Number of Test Cases in the Docstring**    Including test cases in inputs adds execution hints of the expected functionality of the solution, and hence may improve execution accuracy. We test this hypothesis by experimenting with prompts that have varying numbers of test cases. Besides the default setting with zero tests, we compare adding one random test case and all annotated test cases.

Figure 11 (right) shows that injecting as few as one exemplar test case significantly improves the execution accuracy, yet adding more cases has little bonus. This potentially implies the sufficiency of one test case to show the main functionality.

**Number of Evaluation Test Cases**    Execution results could be more reliable if using more test cases for evaluation. However, there is a trade-off between evaluation effectiveness and annotation efficiency, due to the high cost of human effort. To study this tradeoff, we observe how results change with respect to the number of tests. Compared to using all cases in default, we also try using one randomly selected case. For simplicity, we do not include any test cases in prompts.

As shown in Figure 12, evaluating over one random test largely preserves the accuracy of using all tests, indicating that one case is sufficient to test the main functionality for most queries. Check §E for analysis on other factors such as function naming.

## 8    Related Work

**Open Domain Code Generation**    Programs often use APIs from different Python libraries. Some datasets preserve natural coverage from interactive Jupyter Notebooks (Agashe et al., 2019) or Stack-Overflow posts (Yin et al., 2018; Wang et al., 2023),

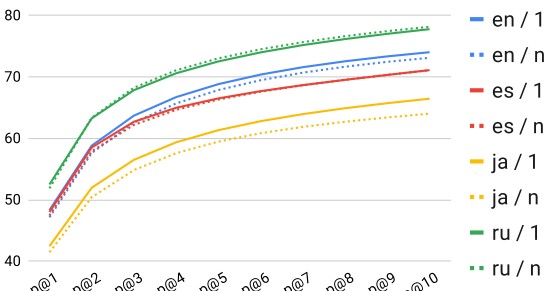

Figure 12: *pass@1* when executing one or all test cases.

but face challenges in enabling execution (Lai et al., 2022; Chandel et al., 2022). Our ODEX dataset addresses execution for open-domain code.

**Coding Queries vs. Programming Challenges**    Some works stem from coding contest websites (Hendrycks et al., 2021; Li et al., 2022), but GitHub Jupyter Notebooks (Agashe et al., 2019; Huang et al., 2022) and StackOverflow (SO) (Yin et al., 2018; Wang et al., 2023; Lai et al., 2022) provide more natural and practical coding queries. We preserve this naturalness and incorporate various NL settings to assist programmers worldwide.

**Execution-based Evaluation**    Evaluation by execution has long been used for SQL (Zhong et al., 2017) or logical forms (Dong and Lapata, 2016). Many datasets have begun to support Python execution via test cases, however focus on built-in functions (Chen et al., 2021; Austin et al., 2021; Hendrycks et al., 2021) or specific domains (Lai et al., 2022; Huang et al., 2022). Our test cases, in contrast, cover diverse libraries in the open domain.

## 9    Conclusion

We present ODEX, an open-domain code generation dataset supporting execution-based evaluation via human-written test cases. ODEX not only supports execution-based evaluation of code using test cases, but also extends the task to the open domain, covering 79 diverse Python libraries and four natural languages (English, Spanish, Japanese, and Russian). Comparing two state-of-the-art code generation models, CODEX and CODEGEN, our dataset effectively unveils their varied behaviors between program domains and language contexts. ODEX serves as a comprehensive NL-to-code benchmark given its open-domain coverage, multi-natural language queries, and multi-metric support. When bringing code execution to open domain scenarios, our explorations also reveal emerging challenges in test creation and reliable execution, which we hope that our dataset will enable future work to tackle.

## Limitations

ODEX aims to serve as a comprehensive testbed, by enabling execution-based evaluation of code in the open domain, with flexible intent inputs in four natural languages. However, we should hold continuous awareness of execution security, multilingual support, and evaluation reliability.

First, execution supports in ODEX enables more rigorous evaluations than other execution-free methods. However, due to the increased complexity of open-domain codes, more inspections are required for execution safety, either for code solutions or test cases. We should always keep alert to avoid concealing malicious code (Wallace et al., 2021) or generating code with security vulnerabilities (Verdi et al., 2020; Pearce et al., 2021).

Second, in addition to English inputs, ODEX also includes intents specified in three other languages. Still, its language coverage is bounded by the available forums in StackOverflow. We hope our initiative can highlight the multilingual nature of program developers, encourage the emergence of similar data resources in other languages, and continuously promote AI programming assistance in languages worldwide.

Third, as ODEX covers wide-ranging code queries in the open domain, it is more suitable for less resource-demanding scenarios such as downstream evaluation or few-shot learning. Although ODEX is larger than many previous datasets with human-written test cases, it is still limited due to the intense human effort required by the curation process. Regarding this, we encourage users of the dataset to conduct significance testing (Dror et al., 2018) and report more substantial model improvements.

## Ethics Statement

Our work has received IRB approval and is licensed under a Creative Commons Attribution-ShareAlike (CC BY-SA) 4.0 International License. The resulting ODEX dataset is built to serve as a benchmark for open-domain code generation, to further facilitate technological advances in AI programming assistance, meanwhile supporting multiple languages to encourage its universal accessibility.

We strive to ensure high data quality and optimize annotation efficiency. We build the ODEX dataset with natural and practical StackOverflow resources and hire annotators with qualified programming proficiency. We provide our annotators with clearly documented instructions, flexible annotation interfaces (Google Sheets, Jupyter Notebooks), and self-verification tools. We (authors) conduct pilot annotation to confirm the clarity of annotation standards and feasibility of the annotation task. We conduct posthoc examinations on the annotation results, both manually and automatically, to obtain assured data quality (100% pass rate).

We respect the contribution and privacy of our annotators. We offer competitive remuneration for their annotation job and treat each one of them fairly. All annotators possess the right to withdraw at any time. We secure that all their personal information is removed before public release.

We conduct systematic analysis from multiple perspectives in the paper, in an attempt to foster public awareness on generating and evaluating programs in the open domain, both in encouraging more advances in this direction, and raising more concerns about the robustness and security of such unique coding problems.

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

# A  ODEX Dataset

## A.1  Library Distribution Statistics

Aside from the illustrations in § 3.1, we list out the detailed statistics of libraries in ODEX, the eight comparison datasets, and the approximated natural distribution.

**ODEX Domain Statistics**  Table 5 lists the number and percentage of occurrences for each library in the ODEX dataset.

| ODEX | | | | | |
|---|---|---|---|---|---|
| **Library** | **Count** | **Frequency** | **Library** | **Count** | **Frequency** |
| none | 440 | 41.90 | functools | 2 | 0.19 |
| pandas | 81 | 7.71 | http | 2 | 0.19 |
| numpy | 80 | 7.62 | obspy | 2 | 0.19 |
| re | 62 | 5.90 | pickle | 2 | 0.19 |
| os | 42 | 4.00 | pytz | 2 | 0.19 |
| collections | 26 | 2.48 | seaborn | 2 | 0.19 |
| matplotlib | 22 | 2.10 | sqlalchemy | 2 | 0.19 |
| datetime | 21 | 2.00 | statistics | 2 | 0.19 |
| urllib | 19 | 1.81 | string | 2 | 0.19 |
| sys | 17 | 1.62 | xlrd | 2 | 0.19 |
| random | 16 | 1.52 | IPython | 1 | 0.10 |
| io | 15 | 1.43 | argparse | 1 | 0.10 |
| json | 15 | 1.43 | aspose | 1 | 0.10 |
| subprocess | 13 | 1.24 | bisect | 1 | 0.10 |
| requests | 10 | 0.95 | cgi | 1 | 0.10 |
| bs4 | 9 | 0.86 | configparser | 1 | 0.10 |
| itertools | 9 | 0.86 | ctypes | 1 | 0.10 |
| operator | 9 | 0.86 | dateutil | 1 | 0.10 |
| time | 9 | 0.86 | difflib | 1 | 0.10 |
| math | 8 | 0.76 | docxtpl | 1 | 0.10 |
| builtins | 6 | 0.57 | filecmp | 1 | 0.10 |
| selenium | 6 | 0.57 | ftplib | 1 | 0.10 |
| tensorflow | 6 | 0.57 | hashlib | 1 | 0.10 |
| django | 5 | 0.48 | heapq | 1 | 0.10 |
| sqlite3 | 5 | 0.48 | imp | 1 | 0.10 |
| PIL | 4 | 0.38 | inspect | 1 | 0.10 |
| codecs | 4 | 0.38 | locale | 1 | 0.10 |
| cv2 | 4 | 0.38 | lxml | 1 | 0.10 |
| scipy | 4 | 0.38 | mechanize | 1 | 0.10 |
| sklearn | 4 | 0.38 | mpl_toolkits | 1 | 0.10 |
| base64 | 3 | 0.29 | multidict | 1 | 0.10 |
| csv | 3 | 0.29 | pprint | 1 | 0.10 |
| flask | 3 | 0.29 | queue | 1 | 0.10 |
| glob | 3 | 0.29 | regex | 1 | 0.10 |
| shutil | 3 | 0.29 | rsa | 1 | 0.10 |
| socket | 3 | 0.29 | ssl | 1 | 0.10 |
| struct | 3 | 0.29 | texttable | 1 | 0.10 |
| sympy | 3 | 0.29 | unicodedata | 1 | 0.10 |
| xlwt | 3 | 0.29 | warnings | 1 | 0.10 |
| ast | 2 | 0.19 | xml | 1 | 0.10 |

Table 5: ODEX library distribution.

**Domain Statistics of Comparison Datasets**  Table 6 lists the library frequency of eight comparison dataset mentioned in § 3: HumanEval, MBPP, APPS, MTPB, P3, DSP, DS-1000, and Exe-DS.

| HumanEval | | | | | |
|---|---|---|---|---|---|
| **Library** | **Count** | **Frequency** | **Library** | **Count** | **Frequency** |
| none | 155 | 94.51 | | | |
| math | 6 | 3.66 | hashlib | 1 | 0.61 |
| collections | 1 | 0.61 | re | 1 | 0.61 |

| MBPP | | | | | |
|---|---|---|---|---|---|
| **Library** | **Count** | **Frequency** | **Library** | **Count** | **Frequency** |
| none | 794 | 81.52 | | | |
| re | 73 | 7.49 | cmath | 3 | 0.31 |
| math | 37 | 3.80 | operator | 3 | 0.31 |
| collections | 25 | 2.57 | array | 0 | 0.00 |
| heapq | 16 | 1.64 | bisect | 2 | 0.21 |
| itertools | 12 | 1.23 | copy | 1 | 0.10 |
| sys | 7 | 0.72 | datetime | 1 | 0.10 |

| APPS | | | | |
|---|---|---|---|---|
| **Library** | **Count** | **Frequency** | | |
| none | 10,000 | 100.00 | | |

| MTPB | | | | | |
|---|---|---|---|---|---|
| **Library** | **Count** | **Frequency** | **Library** | **Count** | **Frequency** |
| - | 103 | 88.03 | | | |
| pandas | 3 | 2.56 | collections | 1 | 0.85 |
| re | 3 | 2.56 | datetime | 1 | 0.85 |
| numpy | 2 | 1.71 | math | 1 | 0.85 |
| sklearn | 2 | 1.71 | regex | 1 | 0.85 |

| P3 | | | | | |
|---|---|---|---|---|---|
| **Library** | **Count** | **Frequency** | **Library** | **Count** | **Frequency** |
| - | 1581 | 92.19 | | | |
| itertools | 35 | 2.04 | heapq | 15 | 0.87 |
| random | 31 | 1.81 | re | 15 | 0.87 |
| collections | 28 | 1.63 | math | 10 | 0.58 |

| DSP | | | | | |
|---|---|---|---|---|---|
| **Library** | **Count** | **Frequency** | **Library** | **Count** | **Frequency** |
| - | 2034 | 92.79 | | | |
| sklearn | 110 | 5.02 | collections | 8 | 0.36 |
| numpy | 84 | 3.83 | time | 8 | 0.36 |
| matplotlib | 50 | 2.28 | gzip | 4 | 0.18 |
| pandas | 46 | 2.10 | pickle | 4 | 0.18 |
| scipy | 46 | 2.10 | random | 4 | 0.18 |
| math | 16 | 0.73 | csv | 2 | 0.09 |
| numbers | 12 | 0.55 | itertools | 2 | 0.09 |
| utils | 12 | 0.55 | seaborn | 2 | 0.09 |

| DS-1000 | | | | | |
|---|---|---|---|---|---|
| **Library** | **Count** | **Frequency** | **Library** | **Count** | **Frequency** |
| pandas | 291 | 29.10 | scipy | 106 | 10.60 |
| numpy | 220 | 22.00 | pytorch | 68 | 6.80 |
| matplotlib | 155 | 15.50 | tensorflow | 45 | 4.50 |
| sklearn | 115 | 11.50 | | | |

| Exe-DS | | | | | |
|---|---|---|---|---|---|
| **Library** | **Count** | **Frequency** | **Library** | **Count** | **Frequency** |
| none | 379 | 56.23 | | | |
| sklearn | 75 | 11.13 | pylab | 2 | 0.30 |
| pandas | 58 | 8.61 | __future__ | 1 | 0.15 |
| numpy | 53 | 7.86 | arch | 1 | 0.15 |
| matplotlib | 32 | 4.75 | cPickle | 1 | 0.15 |
| scipy | 18 | 2.67 | cofi | 1 | 0.15 |
| seaborn | 15 | 2.23 | csv | 1 | 0.15 |
| math | 7 | 1.04 | datetime | 1 | 0.15 |
| collections | 4 | 0.59 | functools | 1 | 0.15 |
| re | 4 | 0.59 | graphviz | 1 | 0.15 |
| folium | 3 | 0.45 | json | 1 | 0.15 |
| nltk | 3 | 0.45 | mpl_toolkits | 1 | 0.15 |
| statsmodels | 3 | 0.45 | operator | 1 | 0.15 |
| warnings | 3 | 0.45 | os | 1 | 0.15 |
| IPython | 2 | 0.30 | tensorflow | 1 | 0.15 |

Table 6: Library statistics of eight comparison datasets.

**Approximated Natural Domain Distribution**
To approximate the natural distribution of libraries in the open domain, we count the number of Python files on GitHub that imports the library of interest. Following the GitHub search syntax,[8] we use the query `import ${library_name}` to search files that import a certain library, and use `NOT import` to count files not using any libraries. Their frequencies are shown in Table 7.

| Approximated Natural Distribution | | | |
|---|---|---|---|
| **Library** | **Count** | **Library** | **Count** |
| os | 30,188,921 | sqlite3 | 694,794 |
| sys | 24,213,844 | configparser | 640,014 |
| numpy | 20,965,506 | queue | 631,326 |
| re | 11,762,193 | ssl | 602,351 |
| time | 5,946,718 | http | 597,866 |
| pandas | 5,878,651 | xml | 574,030 |
| random | 5,740,444 | seaborn | 567,576 |
| matplotlib | 5,416,874 | imp | 560,862 |
| json | 4,792,536 | builtins | 560,148 |
| tensorflow | 4,720,266 | locale | 542,607 |
| argparse | 4,570,391 | ast | 444,349 |
| subprocess | 4,165,781 | bisect | 315,031 |
| string | 4,114,004 | pytz | 295,167 |
| codecs | 3,973,691 | heapq | 281,393 |
| warnings | 3,824,001 | cgi | 277,852 |
| math | 3,569,158 | unicodedata | 267,310 |
| django | 3,447,092 | regex | 235,800 |
| shutil | 2,999,394 | difflib | 225,154 |
| requests | 2,837,310 | PIL | 218,526 |
| cv2 | 2,575,063 | sklearn | 208,913 |
| datetime | 2,536,970 | statistics | 127,725 |
| socket | 2,489,033 | rsa | 122,447 |
| pickle | 2,419,604 | lxml | 111,742 |
| io | 2,190,998 | dateutil | 107,041 |
| collections | 2,152,651 | bs4 | 90,224 |
| glob | 2,114,567 | xlrd | 86,522 |
| itertools | 1,899,461 | filecmp | 79,328 |
| urllib | 1,809,462 | IPython | 73,274 |
| flask | 1,788,601 | sympy | 70,969 |
| csv | 1,680,232 | selenium | 56,709 |
| functools | 1,433,520 | xlwt | 55,035 |
| pprint | 1,378,679 | ftplib | 52,121 |
| base64 | 1,352,623 | multidict | 29,224 |
| hashlib | 1,330,158 | mechanize | 20,978 |
| scipy | 1,121,371 | obspy | 5,799 |
| inspect | 1,112,770 | texttable | 4,749 |
| operator | 1,104,841 | aspose | 1,048 |
| ctypes | 864,108 | docxtpl | 76 |
| sqlalchemy | 814,096 | mpl_toolkits | 2 |
| struct | 787,484 | | |

Table 7: Approximated natural domain distribution.

## A.2 More Annotation Details

Along with the NL-Code pair, we also provide IDs of the source StackOverflow post, using which annotators can trace back to the original post webpage

---

and get a better understanding of the question. If any errors or under-specification are spotted in the given NL or code, we ask the annotators to correct it by making the minimal change possible.

Aligning with how programmers import a library, we require the expressions be written in three forms: (1) `import ${LIBRARY}`, (2) `import ${LIBRARY} as ${ABBR}`, or (3) `from ${LIBRARY} import ${FUNCTION}`, where the `${LIBRARY}` can also be sub-classes such as `matplotlib.pyplot`.

We encourage the annotators to use the language identical to the given NL intent when creating the test cases, especially if the code involves string-related operations (e.g., writing regular expressions in Japanese). We encourage the annotators to write reasonably more and diverse test cases, by varying the values or types of variables.

Please find the full instruction[9] and examples[10] for annotation in our code repository.

## B Baseline Results

According to the baseline results in § 5.1, we provide more detailed evaluation results, on the execution pass rate ranging from the top-1 to top-10 model predictions. Table 8 and Table 9 show the zero-shot execution accuracy of CODEX and CODE-GEN models, respectively.

| Model | NL | Pass Rate | | | | | | | | | |
|---|---|---|---|---|---|---|---|---|---|---|---|
| | | @1 | @2 | @3 | @4 | @5 | @6 | @7 | @8 | @9 | @10 |
| C1 | en | 31.91 | 44.67 | 51.81 | 56.54 | 59.95 | 62.56 | 64.61 | 66.28 | 67.65 | 68.79 |
| | es | 31.89 | 43.33 | 49.23 | 53.01 | 55.72 | 57.81 | 59.52 | 60.96 | 62.22 | 63.33 |
| | ja | 25.67 | 36.69 | 42.66 | 46.49 | 49.27 | 51.44 | 53.23 | 54.76 | 56.10 | 57.32 |
| | ru | 40.00 | 53.48 | 60.04 | 63.96 | 66.63 | 68.62 | 70.17 | 71.44 | 72.50 | 73.41 |
| D1 | en | 33.62 | 46.65 | 53.27 | 57.34 | 60.18 | 62.31 | 64.00 | 65.37 | 66.49 | 67.43 |
| | es | 36.89 | 49.46 | 55.44 | 58.96 | 61.37 | 63.22 | 64.78 | 66.20 | 67.56 | 68.89 |
| | ja | 31.04 | 42.11 | 47.83 | 51.54 | 54.26 | 56.39 | 58.11 | 59.53 | 60.67 | 61.59 |
| | ru | 43.21 | 57.53 | 63.93 | 67.58 | 70.03 | 71.85 | 73.29 | 74.51 | 75.60 | 76.59 |
| D2 | en | 47.15 | 57.61 | 62.58 | 65.69 | 67.87 | 69.47 | 70.70 | 71.67 | 72.46 | 73.12 |
| | es | 47.44 | 57.90 | 62.20 | 64.65 | 66.33 | 67.61 | 68.65 | 69.53 | 70.33 | 71.11 |
| | ja | 41.46 | 50.42 | 54.84 | 57.59 | 59.47 | 60.84 | 61.87 | 62.71 | 63.41 | 64.02 |
| | ru | 51.87 | 63.36 | 68.25 | 71.09 | 73.03 | 74.5 | 75.67 | 76.64 | 77.46 | 78.17 |

Table 8: CODEX zero-shot performance.

| Model | NL | Pass Rate | | | | | | | | | |
|---|---|---|---|---|---|---|---|---|---|---|---|
| | | @1 | @2 | @3 | @4 | @5 | @6 | @7 | @8 | @9 | @10 |
| 350M | en | 26.26 | 32.18 | 35.46 | 37.59 | 39.10 | 40.22 | 41.08 | 41.78 | 42.35 | 42.82 |
| | es | 16.67 | 21.85 | 24.70 | 26.56 | 27.82 | 28.68 | 29.27 | 29.65 | 29.89 | 30.00 |
| | ja | 17.44 | 22.86 | 25.51 | 27.12 | 28.21 | 28.97 | 29.52 | 29.93 | 30.24 | 30.49 |
| | ru | 25.87 | 31.44 | 34.27 | 36.11 | 37.44 | 38.44 | 39.22 | 39.86 | 40.40 | 40.87 |
| 2.7B | en | 37.74 | 42.58 | 44.92 | 46.36 | 47.36 | 48.11 | 48.70 | 49.18 | 49.57 | 49.89 |
| | es | 36.44 | 40.89 | 42.83 | 44.01 | 44.84 | 45.48 | 45.96 | 46.32 | 46.56 | 46.67 |
| | ja | 31.83 | 35.70 | 37.64 | 38.80 | 39.58 | 40.13 | 40.56 | 40.92 | 41.22 | 41.46 |
| | ru | 45.67 | 49.83 | 52.07 | 53.50 | 54.54 | 55.37 | 56.04 | 56.61 | 57.10 | 57.54 |
| 6.1B | en | 34.49 | 37.91 | 39.55 | 40.52 | 41.18 | 41.69 | 42.11 | 42.47 | 42.78 | 43.05 |
| | es | 28.56 | 32.05 | 33.85 | 35.03 | 35.86 | 36.48 | 36.94 | 37.28 | 37.56 | 37.78 |
| | ja | 35.55 | 40.11 | 42.04 | 43.25 | 44.12 | 44.77 | 45.28 | 45.69 | 46.04 | 46.34 |
| | ru | 44.64 | 47.29 | 48.53 | 49.28 | 49.82 | 50.23 | 50.56 | 50.82 | 51.03 | 51.19 |

Table 9: CODEGEN zero-shot performance.

---

[8]https://docs.github.com/en/search-github/searching-on-github/searching-code

[9]https://anonymous.4open.science/r/odex/data/instruction.md

[10]https://anonymous.4open.science/r/odex/data/sample_annotation.ipynb

## C Domain-Wise Execution Results

We list out detailed results for experiments in §5.

### C.1 Open Domain Versus Closed Domain

Table 10 and Table 11 shows the execution accuracy for CODEX and CODEGEN on open-domain and closed-domain problems, respectively.

| NL | Split | Pass Rate | | | | | | | | | |
|---|---|---|---|---|---|---|---|---|---|---|---|
| | | @1 | @2 | @3 | @4 | @5 | @6 | @7 | @8 | @9 | @10 |
| | | CODE-CUSHMAN-001 | | | | | | | | | |
| en | - | 31.91 | 44.67 | 51.81 | 56.54 | 59.95 | 62.56 | 64.61 | 66.28 | 67.65 | 68.79 |
| | open | 24.39 | 35.82 | 43.08 | 48.22 | 52.04 | 54.97 | 57.27 | 59.10 | 60.57 | 61.74 |
| | close | 40.19 | 54.42 | 61.41 | 65.69 | 68.66 | 70.90 | 72.70 | 74.18 | 75.45 | 76.56 |
| es | - | 31.89 | 43.33 | 49.23 | 53.01 | 55.72 | 57.81 | 59.52 | 60.96 | 62.22 | 63.33 |
| | open | 27.71 | 38.98 | 45.12 | 49.14 | 52.06 | 54.34 | 56.20 | 57.78 | 59.17 | 60.42 |
| | close | 36.67 | 48.31 | 53.93 | 57.44 | 59.91 | 61.79 | 63.31 | 64.60 | 65.71 | 66.67 |
| ja | - | 25.67 | 36.69 | 42.66 | 46.49 | 49.27 | 51.44 | 53.23 | 54.76 | 56.10 | 57.32 |
| | open | 21.24 | 30.29 | 35.16 | 38.34 | 40.71 | 42.61 | 44.20 | 45.55 | 46.73 | 47.79 |
| | close | 35.49 | 50.89 | 59.28 | 64.56 | 68.23 | 71.01 | 73.25 | 75.16 | 76.86 | 78.43 |
| ru | - | 31.91 | 44.67 | 51.81 | 56.54 | 59.95 | 62.56 | 64.61 | 66.28 | 67.65 | 68.79 |
| | open | 25.96 | 36.80 | 42.57 | 46.22 | 48.79 | 50.76 | 52.38 | 53.76 | 55.00 | 56.14 |
| | close | 51.59 | 67.26 | 74.47 | 78.61 | 81.37 | 83.37 | 84.87 | 86.04 | 86.96 | 87.68 |
| | | CODE-DAVINCI-001 | | | | | | | | | |
| en | - | 33.62 | 46.65 | 53.27 | 57.34 | 60.18 | 62.31 | 64.00 | 65.37 | 66.49 | 67.43 |
| | open | 26.91 | 39.25 | 45.97 | 50.25 | 53.33 | 55.70 | 57.62 | 59.21 | 60.57 | 61.74 |
| | close | 41.00 | 54.79 | 61.32 | 65.14 | 67.71 | 69.59 | 71.02 | 72.14 | 73.01 | 73.68 |
| es | - | 36.89 | 49.46 | 55.44 | 58.96 | 61.37 | 63.22 | 64.78 | 66.20 | 67.56 | 68.89 |
| | open | 31.67 | 44.63 | 51.11 | 54.78 | 57.07 | 58.63 | 59.81 | 60.79 | 61.67 | 62.50 |
| | close | 42.86 | 54.97 | 60.40 | 63.73 | 66.28 | 68.46 | 70.46 | 72.38 | 74.29 | 76.19 |
| ja | - | 31.04 | 42.11 | 47.83 | 51.54 | 54.26 | 56.39 | 58.11 | 59.53 | 60.67 | 61.59 |
| | open | 23.72 | 32.72 | 37.88 | 41.48 | 44.21 | 46.36 | 48.08 | 49.46 | 50.53 | 51.33 |
| | close | 47.25 | 62.92 | 69.89 | 73.85 | 76.54 | 78.62 | 80.34 | 81.83 | 83.14 | 84.31 |
| ru | - | 43.21 | 57.53 | 63.93 | 67.58 | 70.03 | 71.85 | 73.29 | 74.51 | 75.60 | 76.59 |
| | open | 28.86 | 41.01 | 47.05 | 50.77 | 53.47 | 55.65 | 57.53 | 59.22 | 60.79 | 62.28 |
| | close | 55.07 | 71.18 | 77.87 | 81.47 | 83.71 | 85.22 | 86.32 | 87.15 | 87.83 | 88.41 |
| | | CODE-DAVINCI-002 | | | | | | | | | |
| en | - | 47.15 | 57.61 | 62.58 | 65.69 | 67.87 | 69.47 | 70.70 | 71.67 | 72.46 | 73.12 |
| | open | 37.52 | 47.52 | 52.81 | 56.32 | 58.86 | 60.79 | 62.29 | 63.48 | 64.43 | 65.22 |
| | close | 57.75 | 68.72 | 73.33 | 76.02 | 77.78 | 79.03 | 79.96 | 80.69 | 81.29 | 81.82 |
| es | - | 47.44 | 57.90 | 62.20 | 64.65 | 66.33 | 67.61 | 68.65 | 69.53 | 70.33 | 71.11 |
| | open | 45.42 | 56.02 | 60.17 | 62.68 | 64.59 | 66.17 | 67.52 | 68.70 | 69.79 | 70.83 |
| | close | 49.76 | 60.05 | 64.52 | 66.89 | 68.32 | 69.26 | 69.94 | 70.48 | 70.95 | 71.43 |
| ja | - | 41.46 | 50.42 | 54.84 | 57.59 | 59.47 | 60.84 | 61.87 | 62.71 | 63.41 | 64.02 |
| | open | 29.47 | 37.70 | 41.91 | 44.59 | 46.44 | 47.75 | 48.72 | 49.44 | 50.00 | 50.44 |
| | close | 68.04 | 78.61 | 83.48 | 86.40 | 88.36 | 89.82 | 91.03 | 92.11 | 93.14 | 94.12 |
| ru | - | 51.87 | 63.36 | 68.25 | 71.09 | 73.03 | 74.5 | 75.67 | 76.64 | 77.46 | 78.17 |
| | open | 34.74 | 46.20 | 51.46 | 54.65 | 56.93 | 58.75 | 60.29 | 61.66 | 62.89 | 64.04 |
| | close | 66.01 | 77.54 | 82.11 | 84.67 | 86.34 | 87.52 | 88.38 | 89.02 | 89.49 | 89.86 |

Table 10: CODEX pass rate in open and closed domains.

| NL | Split | Pass Rate | | | | | | | | | |
|---|---|---|---|---|---|---|---|---|---|---|---|
| | | @1 | @2 | @3 | @4 | @5 | @6 | @7 | @8 | @9 | @10 |
| | | 350M | | | | | | | | | |
| en | - | 26.26 | 32.18 | 35.46 | 37.59 | 39.10 | 40.22 | 41.08 | 41.78 | 42.35 | 42.82 |
| | open | 22.35 | 27.04 | 29.75 | 31.58 | 32.93 | 33.97 | 34.80 | 35.48 | 36.04 | 36.52 |
| | close | 30.57 | 37.84 | 41.75 | 44.21 | 45.89 | 47.09 | 48.00 | 48.71 | 49.28 | 49.76 |
| es | - | 16.67 | 21.85 | 24.70 | 26.56 | 27.82 | 28.68 | 29.27 | 29.65 | 29.89 | 30.00 |
| | open | 16.04 | 19.58 | 21.23 | 22.12 | 22.59 | 22.82 | 22.90 | 22.92 | 22.92 | 22.92 |
| | close | 17.38 | 24.44 | 28.67 | 31.62 | 33.79 | 35.39 | 36.55 | 37.35 | 37.86 | 38.10 |
| ja | - | 17.44 | 22.86 | 25.51 | 27.12 | 28.21 | 28.97 | 29.52 | 29.93 | 30.24 | 30.49 |
| | open | 15.40 | 19.67 | 21.67 | 22.91 | 23.78 | 24.41 | 24.88 | 25.23 | 25.49 | 25.66 |
| | close | 21.96 | 29.93 | 34.04 | 36.46 | 38.02 | 39.07 | 39.80 | 40.35 | 40.78 | 41.18 |
| ru | - | 25.87 | 31.44 | 34.27 | 36.11 | 37.44 | 38.44 | 39.22 | 39.86 | 40.40 | 40.87 |
| | open | 20.53 | 24.89 | 26.83 | 28.12 | 29.08 | 29.81 | 30.38 | 30.84 | 31.23 | 31.58 |
| | close | 30.29 | 36.84 | 40.41 | 42.71 | 44.34 | 45.57 | 46.53 | 47.31 | 47.97 | 48.55 |
| | | 2.7B | | | | | | | | | |
| en | - | 35.24 | 42.87 | 46.75 | 49.11 | 50.68 | 51.78 | 52.59 | 53.19 | 53.64 | 53.99 |
| | open | 26.04 | 33.02 | 36.92 | 39.36 | 41.01 | 42.20 | 43.10 | 43.80 | 44.35 | 44.78 |
| | close | 45.36 | 53.69 | 57.58 | 59.85 | 61.32 | 62.33 | 63.03 | 63.53 | 63.88 | 64.11 |
| es | - | 26.00 | 33.65 | 37.74 | 40.06 | 41.52 | 42.58 | 43.44 | 44.20 | 44.89 | 45.56 |
| | open | 22.50 | 27.45 | 30.68 | 32.96 | 34.76 | 36.32 | 37.76 | 39.12 | 40.42 | 41.67 |
| | close | 30.00 | 40.74 | 45.81 | 48.17 | 49.25 | 49.74 | 49.94 | 50.00 | 50.00 | 50.00 |
| ja | - | 24.27 | 32.10 | 36.45 | 39.22 | 41.13 | 42.51 | 43.54 | 44.30 | 44.82 | 45.12 |
| | open | 18.67 | 23.93 | 26.94 | 28.97 | 30.45 | 31.58 | 32.44 | 33.06 | 33.45 | 33.63 |
| | close | 36.67 | 50.20 | 57.52 | 61.93 | 64.78 | 66.73 | 68.14 | 69.19 | 70.00 | 790.59 |
| ru | - | 39.64 | 48.11 | 52.46 | 55.25 | 57.23 | 58.71 | 59.84 | 60.71 | 61.39 | 61.90 |
| | open | 27.02 | 34.72 | 38.61 | 41.12 | 42.96 | 44.41 | 45.59 | 46.59 | 47.46 | 48.25 |
| | close | 50.07 | 59.18 | 63.90 | 66.93 | 69.03 | 70.53 | 71.61 | 72.38 | 72.90 | 73.19 |
| | | 6.1B | | | | | | | | | |
| en | - | 34.49 | 37.91 | 39.55 | 40.52 | 41.18 | 41.69 | 42.11 | 42.47 | 42.78 | 43.05 |
| | open | 28.30 | 31.57 | 33.21 | 34.25 | 35.02 | 35.64 | 36.17 | 36.64 | 37.04 | 37.39 |
| | close | 41.29 | 44.89 | 46.53 | 47.42 | 47.97 | 48.35 | 48.64 | 48.88 | 49.09 | 49.28 |
| es | - | 28.56 | 32.05 | 33.85 | 35.03 | 35.86 | 36.48 | 36.94 | 37.28 | 37.56 | 37.78 |
| | open | 25.83 | 28.61 | 30.16 | 31.25 | 32.06 | 32.64 | 33.02 | 33.24 | 33.33 | 33.33 |
| | close | 31.67 | 35.98 | 38.08 | 39.35 | 40.21 | 40.86 | 41.41 | 41.90 | 42.38 | 42.86 |
| ja | - | 35.55 | 40.11 | 42.04 | 43.25 | 44.12 | 44.77 | 45.28 | 45.69 | 46.04 | 46.34 |
| | open | 28.76 | 31.96 | 33.36 | 34.23 | 34.83 | 35.28 | 35.62 | 35.89 | 36.11 | 36.28 |
| | close | 50.59 | 58.17 | 61.26 | 63.26 | 64.71 | 65.81 | 66.68 | 67.41 | 68.04 | 68.63 |
| ru | - | 44.64 | 47.29 | 48.53 | 49.28 | 49.82 | 50.23 | 50.56 | 50.82 | 51.03 | 51.19 |
| | open | 28.33 | 30.14 | 31.16 | 31.92 | 32.53 | 33.04 | 33.45 | 33.78 | 34.04 | 34.21 |
| | close | 58.12 | 61.47 | 62.87 | 63.63 | 64.10 | 64.43 | 64.69 | 64.90 | 65.07 | 65.22 |

Table 11: CODEGEN pass rate in various domains.

### C.2 Domain-wise Execution Accuracy

As introduced in § 5.3, we take CODE-DAVINCI-002, and report its execution accuracy on each domain in Table 12.

| Library | Count | Pass@1 | Library | Count | Pass@1 |
|---|---|---|---|---|---|
| none | 440 | 61.45 | functools | 2 | 15.00 |
| pandas | 81 | 38.52 | http | 2 | 40.00 |
| numpy | 80 | 36.18 | obspy | 2 | 0.00 |
| re | 62 | 36.13 | pickle | 2 | 0.00 |
| os | 42 | 42.62 | pytz | 2 | 20.00 |
| collections | 26 | 35.38 | seaborn | 2 | 0.00 |
| matplotlib | 22 | 9.00 | sqlalchemy | 2 | 50.00 |
| datetime | 21 | 30.95 | statistics | 2 | 40.00 |
| urllib | 19 | 14.74 | string | 2 | 0.00 |
| sys | 17 | 15.88 | xlrd | 2 | 30.00 |
| random | 16 | 62.00 | IPython | 1 | 0.00 |
| io | 15 | 32.67 | argparse | 1 | 100.00 |
| json | 15 | 35.33 | aspose | 1 | 10.00 |
| subprocess | 13 | 30.77 | bisect | 1 | 0.00 |
| requests | 10 | 37.00 | cgi | 1 | 80.00 |
| bs4 | 9 | 38.89 | configparser | 1 | 60.00 |
| itertools | 9 | 27.78 | ctypes | 1 | 60.00 |
| operator | 9 | 64.44 | dateutil | 1 | 30.00 |
| time | 9 | 20.00 | difflib | 1 | 0.00 |
| math | 8 | 61.43 | docxtpl | 1 | 10.00 |
| builtins | 6 | 76.67 | filecmp | 1 | 40.00 |
| selenium | 6 | 50.00 | ftplib | 1 | 60.00 |
| tensorflow | 6 | 6.67 | hashlib | 1 | 0.00 |
| django | 5 | 20.00 | heapq | 1 | 0.00 |
| sqlite3 | 5 | 38.00 | imp | 1 | 40.00 |
| PIL | 4 | 35.00 | inspect | 1 | 0.00 |
| codecs | 4 | 72.50 | locale | 1 | 0.10 |
| cv2 | 4 | 22.50 | lxml | 1 | 0.00 |
| scipy | 4 | 5.00 | mechanize | 1 | 0.00 |
| sklearn | 4 | 0.00 | mpl_toolkits | 1 | 0.00 |
| base64 | 3 | 6.67 | multidict | 1 | 90.00 |
| csv | 3 | 36.67 | pprint | 1 | 20.00 |
| flask | 3 | 50.00 | queue | 1 | 0.00 |
| glob | 3 | 43.33 | regex | 1 | 100.00 |
| shutil | 3 | 60.00 | rsa | 1 | 10.00 |
| socket | 3 | 40.00 | ssl | 1 | 0.00 |
| struct | 3 | 16.67 | texttable | 1 | 60.00 |
| sympy | 3 | 0.00 | unicodedata | 1 | 90.00 |
| xlwt | 3 | 20.00 | warnings | 1 | 70.00 |
| ast | 2 | 50.00 | xml | 1 | 0.00 |

Table 12: CODE-DAVINCI-001 execution accuracy on each domain subset inside ODEX.

### C.3 Qualitative Error Analysis

To provide more intuitive explanations of the domain divergence aforementioned, we conduct error analysis over 60 randomly selected examples from ODEX dataset (15 for each language). By examining the error patterns from these examples, we aim to answer: what are the common error types on open- and closed-domain problems? What are the main differences between them?

Similar to the previous section, we take the CODE-DAVINCI-002 since it scores the best and presents clear domain gaps, which might give more intuitive variances between domains.

**Closed-Domain Errors** Of the 60 random samples we analyzed, 31 are closed-domain problems, and CODEX predicts erroneous code solutions for 22 of them. We identify four main types of errors

from these samples: (1) 11 cases (50.0%) use the Python built-in functions incorrectly, mostly about strings manipulations and number calculations; (2) 7 cases (31.8%) failed at complex functions, which usually require multi-step implementations; (3) 4 cases (18.2%) received empty predictions, potentially because they involve unfamiliar topics to the model; (4) 2 cases (9.1%) imports extra library or add redundant implementations.

Note that the number of error cases in these four categories does not add up to 22. Since we analyze all of the error predictions among the model top-10 predictions, one case could present multiple error types in its different predictions.

**Open-Domain Errors**   Of the other 29 problems belonging to the open domain, 26 of them have erroneous predictions. Errors in the open domain exhibit more diversity than in the closed domain. The major error enclosing 16 cases (61.5%) is the failure to use the prerequisite libraries, or missing part of them when multiple libraries are involved. The next major type is using incorrect functions, which happens in 9 cases (34.6%). Similarly to the closed-domain errors, 5 cases (19.2%) have error usage of correct functions, 4 cases (15.4%) struggle with complex multi-step implementations, and 3 cases (11.5%) face empty predictions.

OD and CD problems share some error categories such as function misuse and complex operations. Nonetheless, open-domain problems introduce extra challenges: correct selection and usage of libraries and functions in the wild.

# D   Evaluation Metrics

We describe each of the non-execution metrics (§ D.1) as introduced in § 6, report model performance with each (§ D.2), and visualize their correlations with the execution accuracy (§ D.3).

## D.1   Metric Description

**BLEU**   BLEU (Papineni et al., 2002) is a lexical-based evaluation metric, which calculates the n-gram overlap between text prediction and (multiple) references. Most default calculation processes calculate up to 4-grams and adopt the smoothing function introduced in Lin and Och (2004).

**ROUGE**   ROUGE (Lin, 2004) is another more recall-oriented lexical-based evaluation metric. It was originally designed for measuring text summarization, mainly by counting the number of

overlapping units (n-gram, word sequences, and word pairs) between prediction and references. Among the multiple variants proposed (ROUGE-N, ROUGE-L, ROUGE-W, and ROUGE-S), we use the most common ROUGE-L in our experiments.

**METEOR**   METEOR (Banerjee and Lavie, 2005) is a unigram-based metric originally intended for machine translation. It builds on a generalized unigram concept by involving unigram precision, unigram recall, and word order measures.

**ChrF**   ChrF (Popović, 2015) targets lexical match on the character level, by calculating the character-level n-gram F-score between predictions and references. ChrF is also originally proposed for the machine translation task, but later adopted for some code evaluation works (Evtikhiev et al., 2023).

**CodeBLEU**   CodeBLEU (Ren et al., 2020) is specifically designed for code evaluation, by jointly considering the surface-form match, syntax similarly, and semantic data flows.

## D.2   Evaluating with Non-execution Metrics

Table 13 and Table 14 shows the scores of CODEX and CODEGEN using non-execution metrics.

| Model | NL | Metrics | | | | |
|---|---|---|---|---|---|---|
| | | BLEU | ROUGE | METEOR | ChrF | CodeBLEU |
| C1 | en | 31.27 | 52.79 | 55.43 | 43.07 | 3.18 |
| | es | 13.69 | 38.29 | 40.86 | 21.17 | 3.96 |
| | ja | 18.57 | 46.67 | 48.76 | 34.89 | 3.63 |
| | ru | 14.42 | 41.49 | 45.53 | 34.63 | 2.70 |
| D1 | en | 30.94 | 53.88 | 56.01 | 43.60 | 3.27 |
| | es | 20.40 | 43.93 | 46.71 | 29.36 | 3.27 |
| | ja | 19.98 | 48.23 | 51.46 | 38.41 | 3.40 |
| | ru | 16.97 | 44.71 | 47.11 | 35.54 | 2.74 |
| D2 | en | 38.75 | 56.05 | 55.39 | 44.40 | 3.77 |
| | es | 18.47 | 44.98 | 43.52 | 27.11 | 5.78 |
| | ja | 27.10 | 52.04 | 50.17 | 40.02 | 3.58 |
| | ru | 25.00 | 50.04 | 50.51 | 38.60 | 3.75 |

Table 13: CODEX results on non-execution metrics.

| Model | NL | Metrics | | | | |
|---|---|---|---|---|---|---|
| | | BLEU | ROUGE | METEOR | ChrF | CodeBLEU |
| 350M | en | 12.04 | 50.94 | 50.46 | 30.12 | 4.90 |
| | es | 9.07 | 37.90 | 37.76 | 20.90 | 5.47 |
| | ja | 9.43 | 44.21 | 41.29 | 26.16 | 6.05 |
| | ru | 13.35 | 44.77 | 44.27 | 32.40 | 3.86 |
| 2.7B | en | 18.22 | 54.82 | 54.32 | 34.98 | 5.30 |
| | es | 13.05 | 39.79 | 40.93 | 22.61 | 6.67 |
| | ja | 14.72 | 52.46 | 51.22 | 31.28 | 5.42 |
| | ru | 23.27 | 50.82 | 49.98 | 37.75 | 4.31 |
| 6.1B | en | 12.41 | 52.82 | 54.03 | 31.38 | 4.51 |
| | es | 11.69 | 33.26 | 34.47 | 19.04 | 4.57 |
| | ja | 19.14 | 51.31 | 52.07 | 34.78 | 5.68 |
| | ru | 23.66 | 49.09 | 49.48 | 37.44 | 3.72 |

Table 14: CODEGEN results on non-execution metrics.

## D.3 Visualizing Metric Correlations

Following the discussion in § 6, we visualize the non-execution metric metrics between samples that pass and fail during execution time. All experiments use CODE-DAVINCI-002 predictions for evaluation. Figure 13, Figure 14, Figure 15, Figure 16 illustrates the histogram between passed/failed samples using ROUGE, METEOR, ChrF, and CodeBLEU metrics, respectively.

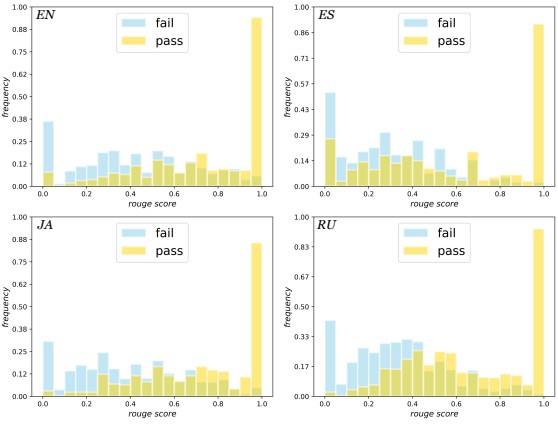

Figure 13: ROUGE on passed and failed samples.

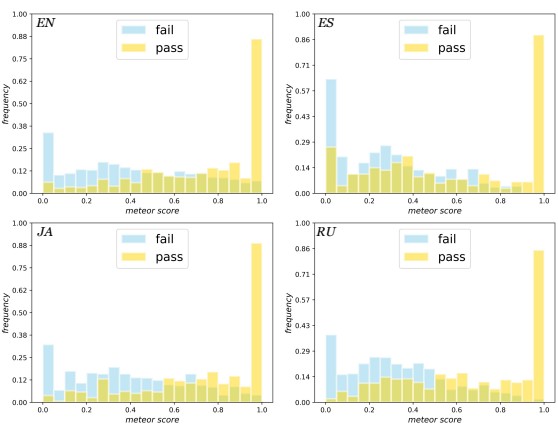

Figure 14: METEOR on passed and failed samples.

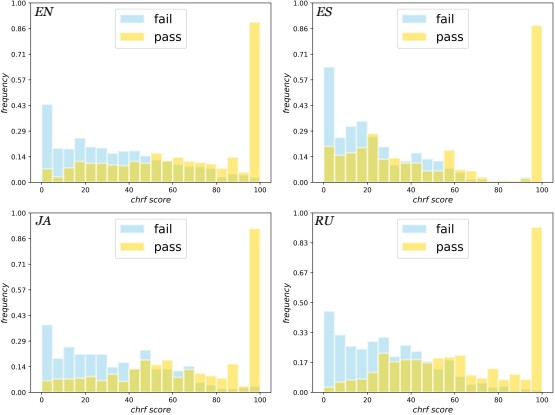

Figure 15: ChrF on passed and failed samples.

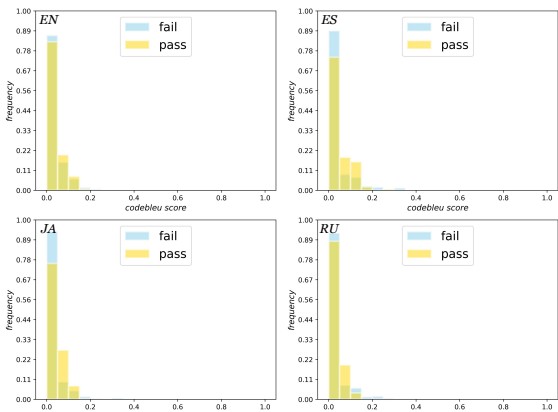

Figure 16: CodeBLEU on passed and failed samples.

## D.4 Why is Execution Better?

To give more intuitive reasons for the advantages of execution, we randomly sample 15 cases from each language subset and identified two major benefits: it tolerates alternative solutions and allows execution results as outputs.

**Alternative Code Implementation** Probably the greatest advantage of execution is it only requires correct execution results, without limitations on alternative methods, as in Figure 17.

```
"""Multiply each value by 2 for all keys in a dictionary `my_dict`"""

# Canonical Solution
my_dict.update((x, y * 2) for x, y in list(my_dict.items()))

# Model Prediction
for key in my_dict:
    my_dict[key] *= 2
```

Figure 17: An alternative yet correct prediction, only has a low $4.8$ BLEU score due to having little lexical overlap with the canonical solution.

**Directly Generating Execution Results** Another interesting category is directly generating the code execution results instead of the implementation steps. This often happens to simple coding queries such as basic string manipulation, where predicting the results might cost the model similar efforts to getting the programmatic solutions.

```
"""Decode a hex string '4a4b4c' to UTF-8."""

# Canonical Solution
bytes.fromhex('4a4b4c').decode('utf-8')

# Model Prediction
'JKL'
```

Figure 18: An example output of a correct execution result, yet only achieving $0.6$ BLEU.

In Figure 18, instead of the string decoding program, the model directly outputs the result string "JLK". While this is somewhat unexpected under

the NL-to-Code task, execution effectively handles such cases and would judge them as correct.

## D.5 Potential Benefit of Lexical-based Metrics

Lexical-based metrics, although relatively ineffective for functional correctness, still are potentially helpful for debugging and interpretation. They are effective in small errors of two types: (1) a single function misuse and (2) slight variance in complex strings. The high lexical match in such cases indicates less effort for fixing (Deng et al., 2021).

**Function Misuse** Some code predictions are correct except for a single place where a wrong function is used, or an argument is misplaced.

```
"""Match regex '\\((.*?)\\)|(\\w)' with string '(zyx)bc'"""

# Canonical Solution
re.findall('\\((.*?)\\)|(\\w)', '(zyx)bc')

# Model Prediction
re.match('\\((.*?)\\)|(\\w)', '(zyx)bc')
```

Figure 19: Example that the model prediction uses the wrong function, having a very high BLEU score 0.925.

For example, in Figure 19, the code imports the library and copies all strings correctly. But it uses the wrong function `match` instead of the correct `findall`. Although the execution fails, the code is similar to the solution. Given the sign of a high BLEU score of 92.5, we could readily spot such similarities and fix them with simple edits.

**String Difference** Another frequent error concerns string copying, where the code calls the correct functions but copies the string differently.

The example in Figure 20 gets a 100.0 BLEU score, but the string inside actually misses a single whitespace, which the BLEU tokenization would discard. Such code also resembles the solution and could be easily fixed by even rule-based methods.

```
"""Initialize `SECRET_KEY` in flask config with 'Your_secret_string '"""

# Canonical Solution
app.config['SECRET_KEY'] = "Your_secret_string "

# Model Prediction
app.config['SECRET_KEY'] = 'Your_secret_string'
```

Figure 20: Example that the model prediction varies slightly in copied strings, but scores 100.0 in BLEU.

## E Ablation Studies

This section provides the results tables according to each ablation study section in § 7.

## E.1 Prompting Strategy

### E.1.1 Few-shot Prompting

Table 15, Table 16, Table 17 show the change in execution accuracy with respect to the examples in in-context learning, on the three CODEX variants

| N-shot | NL | Pass Rate | | | | | | | | | |
|---|---|---|---|---|---|---|---|---|---|---|---|
| | | @1 | @2 | @3 | @4 | @5 | @6 | @7 | @8 | @9 | @10 |
| 1-shot | en | 37.90 | 48.71 | 54.50 | 58.25 | 60.88 | 62.82 | 64.31 | 65.52 | 66.54 | 67.43 |
| | es | 36.22 | 45.51 | 50.70 | 53.96 | 56.14 | 57.68 | 58.81 | 59.70 | 60.44 | 61.11 |
| | ja | 29.76 | 38.54 | 43.22 | 46.23 | 48.33 | 49.90 | 51.14 | 52.15 | 52.99 | 53.66 |
| | ru | 45.67 | 56.75 | 62.32 | 65.86 | 68.38 | 70.32 | 71.88 | 73.21 | 74.37 | 75.40 |
| 2-shot | en | 37.27 | 47.89 | 53.39 | 57.02 | 59.68 | 61.75 | 63.41 | 64.80 | 65.97 | 66.97 |
| | es | 38.56 | 48.77 | 54.12 | 57.50 | 59.90 | 61.75 | 63.26 | 64.54 | 65.67 | 66.67 |
| | ja | 32.26 | 41.57 | 46.71 | 50.18 | 52.76 | 54.78 | 56.40 | 57.74 | 58.84 | 59.76 |
| | ru | 46.75 | 58.56 | 64.24 | 67.63 | 69.90 | 71.55 | 72.82 | 73.84 | 74.68 | 75.40 |
| 3-shot | en | 39.91 | 50.45 | 55.62 | 58.83 | 61.06 | 62.74 | 64.07 | 65.17 | 66.13 | 66.97 |
| | es | 37.00 | 45.88 | 50.05 | 52.63 | 54.48 | 55.87 | 56.95 | 57.80 | 58.44 | 58.89 |
| | ja | 32.87 | 42.48 | 47.58 | 50.88 | 53.29 | 55.16 | 56.66 | 57.89 | 58.90 | 59.76 |
| | ru | 48.33 | 60.03 | 65.32 | 68.51 | 70.71 | 72.35 | 73.66 | 74.75 | 75.71 | 76.59 |

Table 15: CODE-CUSHMAN-001 few-shot results.

| N-shot | NL | Pass Rate | | | | | | | | | |
|---|---|---|---|---|---|---|---|---|---|---|---|
| | | @1 | @2 | @3 | @4 | @5 | @6 | @7 | @8 | @9 | @10 |
| 1-shot | en | 43.05 | 53.67 | 58.80 | 62.01 | 64.31 | 66.09 | 67.52 | 68.71 | 69.73 | 70.62 |
| | es | 41.00 | 52.69 | 58.54 | 62.11 | 64.56 | 66.35 | 67.69 | 68.69 | 69.44 | 70.00 |
| | ja | 35.00 | 45.57 | 51.17 | 54.79 | 57.45 | 59.58 | 61.35 | 62.86 | 64.15 | 65.24 |
| | ru | 47.30 | 59.07 | 64.57 | 67.92 | 70.25 | 72.02 | 73.41 | 74.52 | 75.44 | 76.19 |
| 2-shot | en | 44.26 | 53.98 | 58.77 | 61.85 | 64.00 | 65.59 | 66.79 | 67.70 | 68.43 | 69.02 |
| | es | 40.44 | 50.15 | 54.97 | 57.90 | 59.91 | 61.41 | 62.64 | 63.70 | 64.67 | 65.56 |
| | ja | 35.12 | 44.82 | 49.87 | 53.07 | 55.25 | 56.77 | 57.86 | 58.66 | 59.27 | 59.76 |
| | ru | 49.72 | 60.59 | 65.76 | 68.96 | 71.16 | 72.78 | 74.03 | 75.04 | 75.87 | 76.59 |
| 3-shot | en | 43.58 | 53.27 | 57.88 | 60.81 | 62.99 | 64.74 | 66.19 | 67.44 | 68.52 | 69.48 |
| | es | 41.67 | 53.14 | 58.78 | 62.03 | 64.11 | 65.55 | 66.62 | 67.48 | 68.22 | 68.89 |
| | ja | 38.78 | 49.40 | 54.59 | 57.66 | 59.71 | 61.18 | 62.31 | 63.21 | 63.96 | 64.63 |
| | ru | 49.21 | 58.83 | 63.58 | 66.73 | 69.08 | 70.99 | 72.63 | 74.08 | 75.40 | 76.59 |

Table 16: CODE-DAVINCI-001 few-shot results.

| N-shot | NL | Pass Rate | | | | | | | | | |
|---|---|---|---|---|---|---|---|---|---|---|---|
| | | @1 | @2 | @3 | @4 | @5 | @6 | @7 | @8 | @9 | @10 |
| 1-shot | en | 46.33 | 56.08 | 60.54 | 63.36 | 65.39 | 66.97 | 68.24 | 69.28 | 70.14 | 70.84 |
| | es | 44.33 | 54.00 | 59.04 | 62.49 | 65.07 | 67.09 | 68.72 | 70.07 | 71.22 | 72.22 |
| | ja | 46.33 | 56.08 | 60.54 | 63.36 | 65.39 | 66.97 | 68.24 | 69.28 | 70.14 | 70.84 |
| | ru | 51.35 | 62.72 | 68.20 | 71.60 | 73.98 | 75.79 | 77.24 | 78.47 | 79.56 | 80.56 |
| 2-shot | en | 47.29 | 57.32 | 61.96 | 64.69 | 66.53 | 67.86 | 68.90 | 69.74 | 70.46 | 71.11 |
| | es | 45.78 | 55.85 | 60.41 | 63.29 | 65.44 | 67.16 | 68.63 | 69.93 | 71.11 | 72.22 |
| | ja | 42.38 | 52.28 | 56.80 | 59.38 | 61.02 | 62.12 | 62.88 | 63.41 | 63.78 | 64.02 |
| | ru | 51.75 | 63.38 | 68.47 | 71.51 | 73.60 | 75.13 | 76.30 | 77.23 | 77.98 | 78.57 |
| 3-shot | en | 48.18 | 57.99 | 62.64 | 65.50 | 67.50 | 68.99 | 70.17 | 71.14 | 71.96 | 72.67 |
| | es | 44.44 | 53.95 | 58.31 | 61.07 | 63.11 | 64.74 | 66.07 | 67.19 | 68.11 | 68.89 |
| | ja | 46.10 | 55.64 | 59.74 | 62.17 | 63.81 | 65.00 | 65.90 | 66.61 | 67.20 | 67.68 |
| | ru | 49.40 | 60.56 | 66.09 | 69.64 | 72.19 | 74.17 | 75.77 | 77.12 | 78.29 | 79.37 |

Table 17: CODE-DAVINCI-002 few-shot results.

### E.1.2 Number of Input Test Cases

Table 18 shows the effects on execution accuracy of adding one or more test cases to prompts. Experiments use CODE-DAVINCI-002 as an example.

| # test | NL | Pass Rate | | | | | | | | | |
|---|---|---|---|---|---|---|---|---|---|---|---|
| | | @1 | @2 | @3 | @4 | @5 | @6 | @7 | @8 | @9 | @10 |
| 0 | en | 47.15 | 57.61 | 62.58 | 65.69 | 67.87 | 69.47 | 70.70 | 71.67 | 72.46 | 73.12 |
| | es | 47.44 | 57.90 | 62.20 | 64.65 | 66.33 | 67.61 | 68.65 | 69.53 | 70.33 | 71.11 |
| | ja | 41.46 | 50.42 | 54.84 | 57.59 | 59.47 | 60.84 | 61.87 | 62.71 | 63.41 | 64.02 |
| | ru | 51.87 | 63.36 | 68.25 | 71.09 | 73.03 | 74.5 | 75.67 | 76.64 | 77.46 | 78.17 |
| 1 | en | 63.35 | 75.61 | 80.28 | 82.70 | 84.20 | 85.22 | 85.97 | 86.57 | 87.06 | 87.47 |
| | es | 63.89 | 76.37 | 81.75 | 84.75 | 86.60 | 87.82 | 88.65 | 89.23 | 89.67 | 90.00 |
| | ja | 63.90 | 74.66 | 78.85 | 81.13 | 82.61 | 83.68 | 84.49 | 85.12 | 85.61 | 85.98 |
| | ru | 65.04 | 77.80 | 82.89 | 85.72 | 87.53 | 88.75 | 89.60 | 90.19 | 90.60 | 90.87 |
| n | en | 64.76 | 77.36 | 82.02 | 84.40 | 85.93 | 87.05 | 87.93 | 88.65 | 89.25 | 89.75 |
| | es | 59.89 | 72.42 | 77.44 | 80.41 | 82.49 | 84.03 | 85.16 | 85.95 | 86.44 | 86.67 |
| | ja | 63.41 | 74.02 | 78.49 | 80.98 | 82.57 | 83.69 | 84.51 | 85.14 | 85.61 | 85.98 |
| | ru | 66.67 | 79.07 | 83.70 | 86.19 | 87.82 | 89.01 | 89.91 | 90.62 | 91.19 | 91.67 |

Table 18: CODE-DAVINCI-002 results when using zero (*0*), one (*1*), and all (*n*) test cases in the prompt input.

Furthermore, we experiment on the subset of examples having sufficient test cases, to prevent the

n-test setting being trivialized into the 1-test case. Concretely, we filtered all examples with at least 3 test cases and got 112, 17, 25, and 45 examples in English, Spanish, Japanese, and Russian. Pass@1 results on 0/1/n-test settings are shown in Table 19.

| Language | en | es | ja | ru |
|---|---|---|---|---|
| **0** | 52.7 | 47.1 | 52.0 | 64.4 |
| **1** | 63.4 | 70.6 | 64.0 | 66.7 |
| **n** | 67.9 | 58.8 | 68.0 | 71.1 |

Table 19: Results on examples with 3 or more test cases, using zero (*0*), one (*1*), and all (*n*) test cases.

### E.1.3 Pre-processing: Trailing Whitespaces

While the input construction process may introduce whitespaces at the start and the end of the text sequence, we find CODEGEN model unexpectedly sensitive to trailing whitespaces. As shown in Table 20, removing whitespaces from the prompt input increases the pass rate of all sized CODEGEN models by over 20 percent.

| Model | NL | w/ WS | | | | w/o WS | | | |
|---|---|---|---|---|---|---|---|---|---|
| | | @1 | @2 | @5 | @10 | @1 | @2 | @5 | @10 |
| **350M** | en | 10.32 | 11.29 | 12.24 | 12.53 | 26.26 | 32.18 | 39.10 | 42.82 |
| | es | 17.56 | 17.78 | 17.78 | 17.78 | 16.67 | 21.85 | 27.82 | 30.00 |
| | ja | 7.01 | 8.06 | 9.55 | 10.37 | 17.44 | 22.86 | 28.21 | 30.49 |
| | ru | 21.35 | 24.20 | 26.94 | 28.17 | 25.87 | 31.44 | 37.44 | 40.87 |
| **2B** | en | 14.28 | 15.69 | 16.99 | 17.54 | 37.74 | 42.58 | 47.36 | 49.89 |
| | es | 19.67 | 22.32 | 24.76 | 25.56 | 36.44 | 40.89 | 44.84 | 46.67 |
| | ja | 10.98 | 12.56 | 14.20 | 14.63 | 31.83 | 35.70 | 39.58 | 41.46 |
| | ru | 33.10 | 36.01 | 39.53 | 41.67 | 45.67 | 49.83 | 54.54 | 57.54 |
| **6B** | en | 11.96 | 12.95 | 14.01 | 14.81 | 34.49 | 37.91 | 41.18 | 43.05 |
| | es | 14.78 | 16.64 | 18.70 | 20.00 | 28.56 | 32.05 | 35.86 | 37.78 |
| | ja | 12.44 | 14.34 | 16.51 | 17.68 | 35.55 | 40.11 | 44.12 | 46.34 |
| | ru | 32.86 | 34.45 | 36.28 | 37.30 | 44.64 | 47.29 | 49.82 | 51.19 |

Table 20: CODEGEN results when inputting prompts with and without trailing whitespaces (WS).

We conjecture the gain brought by whitespace stripping to be better distributional alignment with CODEGEN training data. As CODEGEN might be pre-trained on whitespace-stripped text sequences, inputs without whitespaces are potentially more aligned with them, hence resulting in better test-time performance. Meanwhile, note that the tokenization processes for text (natural language) and code (programming language) differ in whitespace-style tokens such as \n or \t. These tokens would be removed by text tokenizers by default, while preserved by code tokenizers since they imply structural information in code pieces.

### E.2 Number of Evaluation Test Cases

Table 21 shows the effect when using different numbers of test cases for execution-based evaluation.

### E.2.1 Number of Evaluation Test Cases

| # test | NL | Pass Rate | | | | | | | | | |
|---|---|---|---|---|---|---|---|---|---|---|---|
| | | @1 | @2 | @3 | @4 | @5 | @6 | @7 | @8 | @9 | @10 |
| **1** | en | 48.31 | 58.81 | 63.70 | 66.72 | 68.83 | 70.39 | 71.59 | 72.55 | 73.35 | 74.03 |
| | es | 48.00 | 58.52 | 62.71 | 64.98 | 66.51 | 67.69 | 68.67 | 69.53 | 70.33 | 71.11 |
| | ja | 42.44 | 51.96 | 56.51 | 59.36 | 61.34 | 62.80 | 63.95 | 64.91 | 65.73 | 66.46 |
| | ru | 52.50 | 63.26 | 67.85 | 70.60 | 72.53 | 74.00 | 75.19 | 76.17 | 77.02 | 77.78 |
| **n** | en | 47.15 | 57.61 | 62.58 | 65.69 | 67.87 | 69.47 | 70.70 | 71.67 | 72.46 | 73.12 |
| | es | 47.44 | 57.90 | 62.20 | 64.65 | 66.33 | 67.61 | 68.65 | 69.53 | 70.33 | 71.11 |
| | ja | 41.46 | 50.42 | 54.84 | 57.59 | 59.47 | 60.84 | 61.87 | 62.71 | 63.41 | 64.02 |
| | ru | 51.87 | 63.36 | 68.25 | 71.09 | 73.03 | 74.50 | 75.67 | 76.64 | 77.46 | 78.17 |

Table 21: CODE-DAVINCI-002 results when using different numbers of test cases for execution-based evaluation. *1* means using one randomly selected test case, *n* means using all annotated test cases in ODEX.

We also evaluate on the subset of examples having at least 3 test cases. Table 22 shows the pass@1 results for each language.

| Language | en | es | ja | ru |
|---|---|---|---|---|
| **1** | 60.7 | 64.7 | 68.0 | 73.3 |
| **n** | 52.7 | 47.1 | 52.0 | 64.4 |

Table 22: Results on examples with 3 or more test cases, using one (*1*) or all (*n*) test cases at evaluation.

### E.3 Semantics of Function Names

Because code is wrapped into functions to enable execution, how functions are named may affect model predictions. By default, we name functions using the post ID (e.g., f_3844801), which expresses little semantics of queries. So we try two other methods: (1) a constant string function; and (2) summary phrases from NL intents, e.g., find_max_value.

To do (2), we conduct a heuristic phrase extraction. We first cut the NL intent into words by whitespace, then remove the stop words ('in', 'of', 'a', 'to', 'and', 'for', 'with', 'that') and meaningless punctuations, lastly, concatenate the first $M = 4$ words with '_'. For example, given an intent "decode a hex string '4a4b4c' to UTF-8", the resulting function name would be "decode_a_hex_string". However, for languages that do not separate words with whitespace, this approach may produce less meaningful strings, hence contributing to the inferior performance as shown below.

To fairly compare with previous results, we do not add test cases in prompts.

From Figure 21 and Table 23, using more semantically meaningful functional names barely improves over the default setting. Intuitively, summarizing names from intents adds no extra semantics, but may cost information loss at the curation step, both contributing to the performance drop.

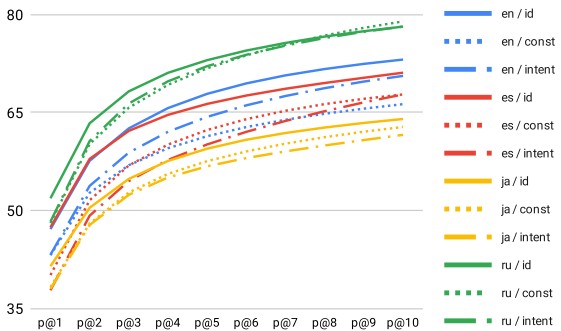

Figure 21: *pass@1* using different function names.

| Func Name | NL | Pass Rate | | | | | | | | | |
|---|---|---|---|---|---|---|---|---|---|---|---|
| | | @1 | @2 | @3 | @4 | @5 | @6 | @7 | @8 | @9 | @10 |
| task-id | en | 47.15 | 57.61 | 62.58 | 65.69 | 67.87 | 69.47 | 70.70 | 71.67 | 72.46 | 73.12 |
| | es | 47.44 | 57.90 | 62.20 | 64.65 | 66.33 | 67.61 | 68.65 | 69.53 | 70.33 | 71.11 |
| | ja | 41.46 | 50.42 | 54.84 | 57.59 | 59.47 | 60.84 | 61.87 | 62.71 | 63.41 | 64.02 |
| | ru | 51.87 | 63.36 | 68.25 | 71.09 | 73.03 | 74.50 | 75.67 | 76.64 | 77.46 | 78.17 |
| constant | en | 43.14 | 52.71 | 56.94 | 59.54 | 61.38 | 62.78 | 63.90 | 64.82 | 65.60 | 66.29 |
| | es | 40.11 | 51.58 | 56.90 | 60.10 | 62.33 | 63.99 | 65.28 | 66.30 | 67.11 | 67.78 |
| | ja | 38.29 | 48.01 | 52.76 | 55.61 | 57.56 | 59.05 | 60.24 | 61.23 | 62.07 | 62.80 |
| | ru | 48.06 | 60.19 | 65.75 | 69.25 | 71.78 | 73.77 | 75.41 | 76.80 | 77.98 | 78.97 |
| intent | en | 43.23 | 53.77 | 58.87 | 62.06 | 64.34 | 66.11 | 67.54 | 68.74 | 69.75 | 70.62 |
| | es | 37.78 | 49.21 | 54.52 | 57.75 | 60.12 | 62.05 | 63.72 | 65.21 | 66.56 | 67.78 |
| | ja | 37.99 | 47.78 | 52.40 | 55.06 | 56.77 | 58.03 | 59.05 | 59.96 | 60.79 | 61.59 |
| | ru | 48.29 | 60.64 | 66.39 | 69.79 | 72.11 | 73.86 | 75.26 | 76.42 | 77.38 | 78.17 |

Table 23: CODE-DAVINCI-002 results when the wrapping function name contains different semantics.

## F ODEX Results on Additional Models

It is possible that some source StackOverflow (SO) posts used to create ODEX examples were used in the training data of the closed-source models in our experiments. However, we have no way to remove those overlapping examples due to the lack of a detailed web index within the training data of these models. On the one hand, we modified the NL intents and code solutions to some extent §2.2, which may alleviate exact matches to scraped training data and, hence reduce the influence of unqualified training data (Lai et al., 2022).

| Language | Domain | SantaCoder | StarCoderBase | StarCoder |
|---|---|---|---|---|
| en | all | 37.65 | 46.51 | 44.67 |
| | open | 30.87 | 40.65 | 37.00 |
| | closed | 45.12 | 52.97 | 53.11 |
| es | all | 32.11 | 30.11 | 37.56 |
| | open | 26.04 | 25.42 | 32.92 |
| | closed | 39.05 | 35.48 | 42.86 |
| ja | all | 28.11 | 41.22 | 44.21 |
| | open | 23.01 | 37.61 | 39.56 |
| | closed | 39.41 | 49.22 | 54.51 |
| ru | all | 36.87 | 46.11 | 50.40 |
| | open | 22.98 | 34.04 | 33.77 |
| | closed | 48.33 | 56.09 | 64.13 |

Table 24: STARCODER pass@1 results on ODEX, evaluated on all (*all*), open-domain (*open*), and closed-domain (*closed*) examples.

To further address the train-test data overlap issue in Codex and CodeGen models, we addition-ally evaluate the SantaCoder (Allal et al., 2023) and StarCoder (Li et al., 2023) models, which have not been trained on any SO data. Table 24 shows the pass@1 of 16B StarCoder and StarCoderBase models, where both models show significant gaps between open- and closed-domain queries, thanks to the broad domain coverage of ODEX.

## G Related Work

**Open Domain Code Generation** Code written in general-purpose programming languages often uses classes or functions from external libraries. A few datasets for code generation preserve this open-domain nature. The CONCODE (Iyer et al., 2018) dataset tested generation of Java class methods. Later works target Python generation given the interactive context of Jupyter Notebooks (Agashe et al., 2019) or natural language intents from Stack-Overflow posts (Yin et al., 2018; Wang et al., 2023). Despite their natural coverage, enabling open-domain code execution has faced great challenges given its diversity and complexity (Lai et al., 2022; Chandel et al., 2022). To address this issue, our ODEX provides test cases as code execution contexts for evaluation.

**Code Evaluation via Execution** Execution-based evaluation has been long adopted for domain-specific programming languages such as SQL queries (Zhong et al., 2017) or logical forms (Dong and Lapata, 2016). This execution-based paradigm has not been introduced to general-purpose languages until recently by the HumanEval dataset (Chen et al., 2021), where human-written test cases are provided for code execution. Many works afterward follow this approach, but focus more on closed-domain settings (Austin et al., 2021; Hendrycks et al., 2021) or specific libraries of interest (Lai et al., 2022; Huang et al., 2022). Toward broader execution environments, we provide executable test cases for as many as 79 libraries.

**Coding Queries Versus Programming Challenges** Programs from different sources are organized for various purposes. Coding contest websites such as LeetCode[11] and Codeforces[12] have been used to build many code generation benchmarks (Hendrycks et al., 2021; Li et al., 2022). However, they randomly align with how humans program in practical scenarios. To build datasets

---

[11] https://leetcode.com/
[12] https://codeforces.com/

with natural and practical usage of code, many works use GitHub Jupyter Notebooks (Agashe et al., 2019; Huang et al., 2022) and StackOverflow forums (Yin et al., 2018; Wang et al., 2023; Lai et al., 2022) as a source of naturally-occurring code. We remain such naturalness by using StackOverflow posts, but uniquely from forums in various languages to also assist programmers worldwide.

**Test Case Creation** While most benchmarks use Python test cases annotated by human programmers (Chen et al., 2021; Nijkamp et al., 2023; Lai et al., 2022), challenge-style datasets adopt a more direct approach by crawling from the web (Hendrycks et al., 2021; Li et al., 2022). Another thread of work attempts to generate test cases automatically based on the Python grammar (Lukasczyk and Fraser, 2022), but is largely limited to basic Python functions. Some propose to leverage the power of neural LMs (Tufano et al., 2021; Li et al., 2022), even jointly considering solution and test case generation (Chen et al., 2023). However, the quality and diversity of test cases are not robustly ensured. We hence use high-quality human-written test cases for ODEX evaluation.