# OpenReview forum: "Execution-Based Evaluation for Open-Domain Code Generation"
_EMNLP/2023/Conference — EMNLP 2023 Findings_

### Official Review · Reviewer_d6KA · 2023-08-03

**Soundness:** 2

**Excitement:**

2: Mediocre: This paper makes marginal contributions (vs non-contemporaneous work), so I would rather not see it in the conference.

**Missing References:**

N/A

**Paper Topic And Main Contributions:**

The authors’ main contribution is proposing ODEX, a dataset comprising 945 NL-Code pairs, composed of 4 languages: English, Spanish, Japanese, and Russian. The dataset includes 79 open-domain programs and addresses three of the challenges associated with open-domain code execution: irreproducible runs, randomized outputs, and specialized equivalence checks.

Previous solutions are either closed domain (they do not take into account third party libraries), or they cover a very small amount of open domain libraries. In contrast, the authors state that ODEX is more “colorful” because it covers significantly more open domain libraries, in addition to those in the closed domain.

ODEX is tested on both CODEX and CODEGEN.



**Questions For The Authors:**

A. There is not much information about the HumanEval dataset. Is this similar to the OpenAI dataset? What is the difference between the two?

B. The authors mention that they have undergraduate students to write test cases. My concern is that the undergraduate students go to stackoverflow for help writing their test cases and then the data has just come full circle. The authors do not mention much about how the students write the test cases. How do they make sure to prevent this?

C. If these are questions from the internet, the model might already be trained on this data. How can we be sure that these are not in the training set? For instance, the authors mention that they use training resources from stackoverflow. But they then go on to test CODEX, which is essentially Co-Pilot, which is known to have been trained on stackoverflow data. How can you avoid models being tested on data that was already in the training set?

D. In Figure 7, why is Russian so much better than the other languages? It is not even the language with the highest resources in the dataset. On a similar note, why is Japanese so performant for CODEX?

E. Are the problems the authors picked in different languages equivalent? Or do they pick easier tasks for more difficult languages?



**Reasons To Accept:**

The proposed dataset could potentially have better data quality than existing datasets.



**Reasons To Reject:**

1. The novelty of the paper is unclear. The authors claim to use a 'more diverse' open domain library, but it's not clear what this result is compared against. It's difficult to fully understand why current solutions aren't enough, or what problem they are trying to solve.

2. The small size of the dataset (only 945 code pairs, further divided by language into about 400 for English and only 90 for Spanish) leads to doubts about the credibility of the results. Although the authors address this in the Limitations Section, it does not alleviate these concerns.

3. This paper does not provide any table or graph showing the test results of ODEX with the 'Human Eval dataset', and it lacks an explanation of how and why ODEX is effective and different from other solutions.



**Reproducibility:**

3: Could reproduce the results with some difficulty. The settings of parameters are underspecified or subjectively determined; the training/evaluation data are not widely available.

**Reviewer Confidence:**

4: Quite sure. I tried to check the important points carefully. It's unlikely, though conceivable, that I missed something that should affect my ratings.

**Typos Grammar Style And Presentation Improvements:**

Lines 11-12: “…as intents, in English, Spanish, Japanese, and Russian.” - Comma splice, you don’t need it after ‘intents’.
Lines 73-74: “…we observe satisfactory multilingual capabilities, despite that neither model was…” - Grammatically incorrect, should be something like: “…we observe satisfactory multilingual capabilities, despite the fact that neither model was…”.
Line 76: “…face greater yet varied challenges…” - Missing commas, should be: “…face greater, yet varied, challenges…”.
Line 114: “…further proofs and clarifies…” - I’m not sure if this is completely grammatically wrong, but I would use ‘proves’ instead of ‘proofs’.
Lines 239-240: “DS-1000 restricts to code using libraries…” - Grammatically incorrect, should be: “DS-1000 is restricted exclusively to code using libraries…”.
Line 269: “…of SO queries…” - Inconsistency, SO has not been previously defined. What does it mean?
Lines 298-301: “Models are progressively trained on ThePile, BigQuery, and BigPythong datasets are denoted as NL, MULTI, and MONO.” - The first clause of this sentence is not connected to the second clause, makes for an illogical sentence.
Lines 367-368: “Model upgrades do not always reduce the gaps.” - A bit of an awkward, abrupt sentence mid-paragraph.
Line 374: “It is hence imperative…” - Incorrect usage of the word ‘hence’.
Lines 435-436: “We thus explore to few-shot prefixing…” - Grammatically incorrect, you don’t explore ‘to’.
Lines 454-455: “This potentially implies the sufficiency of one test case to show the main functionality.” - Not grammatically incorrect, just awkwardly and a bit confusingly worded.
Line 469: “Check Appendix E for…” - Unprofessionally worded.
Line 484: “…StackOverflow (SO)…” - This acronym was referenced earlier, so it should have been defined earlier.
Line 487: “…naturalness…” - Not technically wrong, just not a frequently used word in English due to the way it sounds (a bit juvenile).
Line 521: “…we should should…” - Typo, repeated word.
Line 524: “…execution supports in ODEX…” - unnecessary plural, should be: “support”.
Lines 529-530: “We should always keep alert to concealing malicious…” - Grammatically incorrect and awkward to read. Perhaps try: “We should always remain vigilant about concealing malicious…”.
Line 562-568: Wrong tense used throughout this portion of the paragraph, would make more sense in the past tense.

---

> ### Author Rebuttal · Authors · 2023-08-28
>
> Thank you for your review, and for recognizing the quality of ODEX!
>
> ### R1. Novelty of ODEX
> The novelty of ODEX is providing a code evaluation benchmark with *diverse domain coverage*, as we illustrate in Section 3 in comparison with other datasets (Table 3).
> ODEX serves as a unique portal to reveal the significant performance gap between open- and closed-domain queries, even with current top-performing Codex and CodeGen models (Section 5). In contrast, the HumanEval dataset only shows that Codex and CodeGen achieve rather comparable performance [1].
> Our work points out and mitigates the current shortcomings of models and evaluation benchmarks (in overlooking more realistic open-domain queries).
>
> R2:  It's difficult to fully understand why current solutions aren't enough, or what problem they are trying to solve.
> We kindly note that in this work, we focus on the creation of a new *dataset* other than a *solution*
>
> ### R2. Small Size of ODEX
> As discussed in L542-L551, significance testing [1] can be performed to address the reliability of model performance (differences). We took the best Codex (davinci-002) and CodeGen (6.1B) models and performed significance tests with the pass@1 predictions using a sample rate of 0.5 and a sample size of 10, 000, and it shows win rates of 1.00 for Codex on all language subsets, with p-value << 0.005. We will perform more comprehensive tests and highlight these results in the revised version.
> We also kindly highlight that our ODEX is among the largest execution-based code evaluation benchmarks, given the difficulty of annotating (i.e., hand-writing) evaluation test cases for code. For instance, HumanEval only has 164 examples while ODEX has 945 test cases.
>
>
> ### R3. Testing ODEX on HumanEval Dataset
> We would like to note that ODEX is a code evaluation dataset, the same as the HumanEval dataset. ODEX is not a proposed method for code generation, therefore cannot be evaluated on datasets. We can, though, test code generation models on ODEX, which we experimented in details in Section 5, 6, 7, and Appendices.
>
> ### Q.A. The “HumanEval Dataset” vs. the “OpenAI dataset”
> We have introduced the HumanEval dataset (line 226), illustrated its domain breakdown (Figure 4, Appendix A.1), showed its construction details (Table 3), and summarized model performance on it (line 303, 354). We believe these would provide sufficient background about one of the baseline datasets for our work.
> As far as we know, there are no datasets named the “OpenAI dataset”. You might be referring to the “HumanEval” dataset, which is released by the organization named “OpenAI”. If so, the “HumanEval dataset” and “OpenAI dataset” point to the same dataset.
>
> ### Q.B. Seeking StackOverflow to Write Test Cases
> It is not problematic to write test cases by referring to StackOverflow posts, as long as the tests can correctly evaluate the functionality of canonical solutions (Appendix A.2).
> We performed rounds of screening and annotation tests and hired all qualified annotators, who demonstrated the ability to write multiple valid test cases for all annotatable examples (section 2.3). We believe there are no annotation quality issues directly caused by their undergrad status, and in fact in preliminary experiments we found that the quality of annotations created by these undergrads exceeded that of alleged “professional” freelancers that we found on Upwork.
> To further ensure the test quality, we (authors – graduate students and professors) performed post-hoc examinations and executed the test cases to ensure their correctness (line 195).
>
> ### Q.C. Possible Use of ODEX Data in Model Training
> It is possible that some source StackOverflow posts used to create ODEX examples are used in training by Codex and/or CodeGen. We would remove those overlapping examples if Codex/CodeGen could provide a detailed web index within their training data, but unfortunately, they do not. Nonetheless, we modify the NL intents and code solutions to some extent (described in Section 2.2), which may alleviate exact matches to scraped training data and, hence can reduce the influence of unqualified training data [3].
>
> To further address the train-test data overlap issue, we also evaluate the StarCoder [4] models, which we know have not been trained on any StackOverflow data. The table below shows the pass@1 of 16B StarCoder and StarCoderBase models, where both models show significant gaps between open- and closed-domain queries, thanks to the broad domain coverage of ODEX. We couldn’t report these results at paper submission time since StarCoder was released concurrently. We will add them to the main paper in future revised versions.
>
> | Model         | EN   | - open | - close | ES   | - open | - close | JA   | - open | - close | RU   | - open | - close |
> |---------------|------|--------|---------|------|--------|---------|------|--------|---------|------|--------|---------|
>
> | StarCoderBase | 46.5 | 40.7   | 53.0    | 30.1 | 25.4   | 35.5    | 41.2 | 37.6   | 49.2    | 46.1 | 34.0   | 56.1    |
> | StarCoder     | 44.7 | 37.0   | 53.1    | 37.6 | 32.9   | 42.9    | 44.2 | 39.6   | 54.5    | 50.4 | 33.8   | 64.1    |
>
>
> ### Q.D. Japanese & Russian Performance
> Performance on different language subsets is largely due to: (1) if models are well-train on data of target languages, and (2) the difficulty of included test examples (Section 5.1). We conjecture that good performance comes from both, although we are not as certain as we would be if information about the training data was public. Similarly, Codex could be well-trained on Japanese data hence performing well at testing time.
>
> ### Q.E. Picking Examples for Different Languages
> We do not intentionally “pick” any examples for (sub-)datasets in different languages. Examples are naturally different between language subsets, because we collect queries from respective StackOverflow forums fulfilled by developers native in different languages. Similarly to the findings of (M)CoNaLa, queries asked in some languages (e.g., Spanish) are harder than others (e.g., Russian), presumably due to differences in the culture of Stack Overflow users in the different communities.
>
> ### Typos
> Thank you for the detailed check! We will revise these typos in the final version.
>
>
> [1] Nijkamp, Erik, et al. "Codegen: An open large language model for code with multi-turn program synthesis." arXiv preprint arXiv:2203.13474 (2022).
> [2] Dror, Rotem, et al. "The hitchhiker’s guide to testing statistical significance in natural language processing." Proceedings of the 56th annual meeting of the association for computational linguistics (volume 1: Long papers). 2018.
> [3] Lai, Yuhang, et al. "DS-1000: A natural and reliable benchmark for data science code generation." International Conference on Machine Learning. PMLR, 2023.
> [4] Li, Raymond, et al. "StarCoder: may the source be with you!." arXiv preprint arXiv:2305.06161 (2023).

---

### Official Review · Reviewer_rKvX · 2023-08-05

**Soundness:** 4

**Excitement:**

4: Strong: This paper deepens the understanding of some phenomenon or lowers the barriers to an existing research direction.

**Paper Topic And Main Contributions:**

This paper presents ODEX, a natural language to Python code dataset, supporting execution-based evaluation of code generation models. The salient features are 1) the dataset being “open-domain” where code could import and use one of 79 diverse libraries, 2) 1707 human-written test cases to support execution-based evaluation, and 3) multilingual queries.

**Questions For The Authors:**

A.	Insufficient analysis of test cases written (the main addition in when creating ODEX). Are these test cases diverse/sufficient/coverage/non-repetitive /etc?

B.	Shouldn’t open-domain code gen models also generate correct library imports? The current setup (line 134 – step2) specifies library prerequisites manually, and the prompt for Codex/Codegen already contains the correct library calls. I’m not sure if this is truly open-domain then.

C.	Error analysis of the code generated by Codex and Codegen in open vs closed domains might be helpful. Do the types of errors change, etc?

D.	How are in-context examples chosen? Randomly / similarity match, etc? Details could be included in the appendix (in-context example selection, prompt design, etc)

E.	Line 071 – augmented training data improves execution accuracy – where are these results?

F.	Any instances where the canonical code solution was wrong and didn’t pass test cases? Were these excluded?

G.	Figure 6 has your zero shot prompt, where is the library call? Does the model have to generate this or is this included by not shown in the header?

H.	Nucleus sampling is used. Line 356 compares pass@1 performance between Codex and Codegen. Due to randomness in the generated sample, these results could change. Was any consistency performed?

**Reasons To Accept:**

1.	The main contribution is human-written test cases to test execution-based evaluation.
2.	ODEX tests “open-domain” code generation, from natural and practical queries, requiring code to import and use diverse libraries. The diversity in libraries included in ODEX compared to other datasets is suitably highlighted in the paper.
3.	The dataset collection and annotation procedure are described in detail. Ideas to address challenges (irreproducible runs, randomized outputs, specialized equivalence checks) in test case creation for certain library calls are insightful.
4.	Interesting result comparing the gap between Codex and Codegen for open vs closed domain as the model size increases.

**Reasons To Reject:**

1.	ODEX is essentially a subset of the existing (M)CoNaLa datasets with the addition of human-written test cases. My main concern is the effectiveness of these test cases for execution-based evaluation to check code correctness reliably. With suitability, I refer to the test cases as being necessary and sufficient.
a.	Sufficient test cases would aim for 100% code coverage and handling edge cases, among other things. Since this might be hard to achieve through external annotation, a proxy could be to encourage diversity in test cases written.
b.	The authors show both the query and canonical code to the annotator to write test cases, this might bias the annotator to write test cases wrt the canonical code solution only. Since the annotators are undergrad students and not experts, the annotators could have been shown only the query, (optionally asked to write M different correct solutions, N different incorrect solutions possibly excluding certain corner cases), and test cases (along with the intention of what is being checked behind each test case). The resulting test cases could be less biased wrt the canonical code solution.
c.	The authors test all test cases are correct in 2.4, but this does not guarantee sufficiency/diversity/suitability/non-repetitive.
d.	Insufficient analysis of test cases written (the main addition in when creating ODEX). Are these test cases diverse/sufficient/coverage /non-repetitive / etc?
2.	The authors show two experiments in Figure 11 right and Figure 12, the first increasing the number of test cases as in-context examples didn’t improve performance, and the second stating that the evaluation metrics didn’t change including additional test cases. These results could show the sufficiency of test cases in the dataset, conversely, the results could also support the insufficiency of test cases in the dataset, i.e., adding the remaining test cases didn’t change performance/evaluation. Further, since the average no of test cases is 1.8, isn’t using 1 vs all test cases, essentially 1 vs 1.8 test cases, which might not be suitable for these experiments. Maybe testing on a subset where the number of all test cases >= 3 or 5 might help.
3.	The number of samples for half(?) of the libraries is 1 or 2 (table 5 of Appendix A). This might not be representative to evaluate code gen performance on that library.

I’m open to increasing my score if the above comments are addressed.

**Reproducibility:**

4: Could mostly reproduce the results, but there may be some variation because of sample variance or minor variations in their interpretation of the protocol or method.

**Reviewer Confidence:**

3: Pretty sure, but there's a chance I missed something. Although I have a good feel for this area in general, I did not carefully check the paper's details, e.g., the math, experimental design, or novelty.

---

> ### Author Rebuttal · Authors · 2023-08-28
>
> Thank you for your review and your positive comments about the quality and diversity of our dataset!
>
> ### R1. Sufficiency of Test Cases
> Evaluating the main functionality of code is a critical first step for open-domain code evaluation. It has already taken us substantial efforts (~ 6 months).
>
> ODEX is challenging and largely effective in judging erroneous predictions. As shown in Table 4, the score of the strongest model is somewhat low
> We acknowledge that test coverage can be a potential issue when the test cases in our dataset are no longer effective. Enhancing the test case sufficiency is an important next step to further improve the comprehensiveness of benchmarks. In the future, we could leverage similar methodologies that are used to create HumanEval+ [1].
>
> ### R1. Biased Tests with Canonical Solutions
> Annotating test cases when knowing certain solutions may bias the test cases. However, our test cases basically consist of input-output data, which should be identical regardless of the specific implementation of the solution. Therefore, knowing the canonical solution should only minorly affect the test cases.
> Also, in practice, it is sometimes impossible to annotate tests without solutions, due to the inherent ambiguity of natural languages. Example solutions can provide the necessary specifications. For example, an NL query “sum the first column” is confusing for indexing, but the code “a.sum(axis=0)” points to the literal first column and not `index=1`. An alternative to encourage test diversity would be to simultaneously write test cases and solutions, which we would consider if not due to time and cost constraints.
>
> ### R2. Sufficiency of Test Cases
> First, we want to clarify that Figure 11 (right) shows results when adding test cases to the *docstring* of the current test example other than to in-context examples (which should be Figure 11 left). Because we do not include any test cases in (the inputs) of in-context examples, the results cannot directly support the sufficiency of test cases.
> Thank you for pointing out another dimension of sufficiency! We filtered all examples with >=3 test cases (112, 17, 25, and 45 in English, Spanish, Japanese, and Russian).
> In the first table, we compare the Codex davinci-002 performance before and after (i) adding in-docstring test cases, and (ii) altering the number of evaluation test cases.
>
> | (i) Number of In-Docstring Test Cases | EN | ES | JA | RU |
> |:---------------------------------:|------|------|------|------|
> | 0 | 52.7 | 47.1 | 52.0 | 64.4 |
> | 1 | 63.4 | 70.6 | 64.0 | 66.7 |
> | n | 67.9 | 58.8 | 68.0 | 71.1 |
>
> Adding test cases into inputs significantly improves model performance. These results suggest that ODEX test cases cover the main functionality of code queries, adding more test cases provides richer functional guidance, hence improving the quality of the generated code.
>
> | (ii) Number of Evaluation Test Cases | EN   | ES   | JA   | RU   |
> |---------------------------------|------|------|------|------|
> | 1                               | 60.7 | 64.7 | 68.0 | 73.3 |
> | n                               | 52.7 | 47.1 | 52.0 | 64.4 |
>
> Evaluating model predictions with more test cases effectively filters out a huge number of (previously) false positive predictions. This demonstrates the sufficiency of test cases for multi-test examples. We will add these results to the revised paper version, to reflect the valuable discussion on test case sufficiency.
>
>
> ### R3. Representation of Rare Libraries
> We have fewer examples for less frequent libraries (Table 5), and increasing the overall number of examples could make the benchmark even more robust. However, to preserve the natural distribution of open-domain libraries, we chose to not intentionally create/collect more examples for minority libraries, instead opting to aim for a dataset that is most similar to the distribution of real-world scenarios (Section 3.1). We did our best to collect the largest number of examples from verified StackOverflow sources so far, i.e., (M)CoNaLa, and hope future works could continue to leverage more resources showing up.
> We acknowledge having more examples of rare libraries will enable more robust model evaluations on those libraries. We will discuss this in the limitation section of the paper.
>
> ### Q.A. Analysis of Test Cases
> We encourage annotators to write diverse test cases. To demonstrate this effort, we further analyze the similarity of input arguments in our test cases. For each example with at least two test cases, we take every test pair and calculate the lexical overlap between their argument values using BLEU and Levenshtein edit distance (item-wise, that ‘1’ can be replaced with ‘happy’ in one step). These test pairs score only 13.3 in BLEU, showing a small lexical overlap. They have an average edit distance of 11.3, indicating a broad coverage among test input values.
>
> ### Q.B. Are Import Statements Included
> As illustrated in Figure 6 and described in Section 4 (the Prompt Design paragraph), we do not include any import statements in the prompts, to prevent disclosing any library hints to models. But to avoid penalizing the models if they don’t generate the imports themselves, we add the necessary imports to the evaluation code. The description (in line 134) may cause confusion as annotators were asked to specify required library imports, we will further clarify this detail in experimental sections.
>
> ### Q.C. Error Analysis
> We performed several error analyses: (1) in Appendix C.3, we analyzed typical error types in the open- and closed domains; (2) in Appendix D.5, we spotted interesting error categories by using different evaluation metrics. In summary, error types differ between models and domains. We will move these to the main paper in the camera-ready version, when more space is allowed.
>
> ### Q.D. Selecting In-Context Examples
> We randomly select examples as in-context examples, because this is simple to implement and requires minimum assumptions of the evaluation data. We will clarify this in the revised paper version.
>
> ### Q.E. Augmenting Training Data Helps
> By “augmented training data” we meant the increased training data for models with larger sizes or in upgraded versions, e.g., the 175B Codex davinci-002 compared to the 12B cushman-001.
> Larger and upgraded models achieve better performance, throughout almost all experiments done in this work. We will refine relevant descriptions and clarify this in the revised paper.
>
> ### Q.F. Solution Cannot Pass Tests
> Solutions of all examples can pass the test cases (line 200). This is guaranteed during the annotation stage, while cases to which no annotators can provide a correct test case are filtered out. Some solutions have observable errors in (M)CoNaLa. We asked annotators to fix the solutions if the NL query could be correctly answered and viable test cases could be created accordingly.
>
> ### Q.G. Library Call in Zero-Shot Prompt
> Zero-shot prompt means we include zero information about other examples, not the current example for testing. Regardless of the number of shots, we always import necessary libraries/functions, as described in our response to Q.B (question B) above.
>
> ### Q.H. Consistency Under Nucleus Sampling
> We ran 5 rounds of experiments on the English subset using CodeGen, under the same setting as the experiment reported in Table 5. We list the results of all experiments in the table below.
>
> | Runs \ pass@k | 1 | 2 | 5 | 10 |
> |---------------|---|---|---|----|
> | 0 | 26.26 | 32.18 | 39.10 | 42.82 |
> | 1 | 25.74 | 31.66 | 38.50 | 42.14 |
> | 2 | 26.65 | 32.57 | 39.52 | 43.33 |
> | 3 | 26.10 | 31.94 | 38.72 | 42.21 |
> | 4 | 26.65 | 32.76 | 39.86 | 43.21 |
> | Mean | 26.28 | 32.22 | 39.14 | 42.74 |
> | Std.Dev. | 0.387 | 0.449 | 0.559 | 0.551 |
>
> The average results of all language subsets are close to the reported result, and the standard deviation values are small. Model results are pretty stable under our experiment configuration, therefore the results reported in Table 5 are reliable. We will add results for other languages and models in the revised paper version.
>
>
> [1] Liu, Jiawei, et al. "Is your code generated by chatgpt really correct? rigorous evaluation of large language models for code generation." arXiv preprint arXiv:2305.01210 (2023).

---

### Official Review · Reviewer_FbaY · 2023-08-05

**Soundness:** 4

**Excitement:**

4: Strong: This paper deepens the understanding of some phenomenon or lowers the barriers to an existing research direction.

**Missing References:**

There is also a highly relevant dataset called CodeContest but the authors didn't mention it. It is from the paper 'Competition-Level Code Generation with AlphaCode'


**Paper Topic And Main Contributions:**

This paper proposes a Python code generation dataset named ODEX for execution-based evaluation. This dataset contains 945 NL-Code pairs in 4 languages,  including 439 samples in English, 202 90 in Spanish, 164 in Japanese, and 252 in Russian, with 1707 human written test cases for execution. In this paper, the authors introduce the annotation steps and quality check for creating such a dataset and they conduct detailed analysis of this dataset from various aspects. They also evaluate the performance of existing LLMs for code generation on ODEX. This is a high-quality dataset and it can benefit the researchers who are interested in this domain.

**Questions For The Authors:**

Did you apply filtering as the first step to the original CoNaLa and (M)CoNaLa datasets? If so, how did you filter out the dataset? These two datasets have thousands of data, why are there only 945 pairs left?


**Reasons To Accept:**

* Execution-based evaluation is known for being more aligned with human preference than n-gram based evaluation. Therefore, this dataset with human-written test cases is a great contribution to the community.

* The authors have analyzed the domain distribution by libraries and data complexity comprehensively. The benchmark experiments are also helpful and could shed light on the evaluation of code generation.

* The process steps of creating this dataset are reasonably designed. They also have the quality check during the annotation steps.


**Reasons To Reject:**

It is not very clear from the paper whether the prompt has included the essential import statements. As this dataset is heavily using libraries, it is very important to avoid such ambiguity.


**Reproducibility:**

3: Could reproduce the results with some difficulty. The settings of parameters are underspecified or subjectively determined; the training/evaluation data are not widely available.

**Reviewer Confidence:**

4: Quite sure. I tried to check the important points carefully. It's unlikely, though conceivable, that I missed something that should affect my ratings.

---

> ### Author Rebuttal · Authors · 2023-08-28
>
> Thank you for your review, and your positive comments about the value of our dataset, analysis, and experimentation!
>
> ### R. Are Import Statements Included
> No. As illustrated in Figure 6 and Line 304-315, we do not include any import statements in the prompts, to prevent disclosing any library hints to models. But to avoid penalizing the models if they don’t generate the imports themselves, we add the necessary imports to the evaluation code. The description (in line 134) may cause confusion as annotators were asked to specify required library imports, we will further clarify this detail in experimental sections.
>
> ### Q. Any Filtering on (M)CoNaLa?
> No, we do not filter out any examples in (M)CoNaLa prior to annotations. Nonetheless, some examples in (M)CoNaLa are not included in ODEX for reasons such as imported libraries or functions being deprecated.
>
> ### Missing Reference
> Thanks for the notice! We cited this paper in related works (line 482). We will mention its name “CodeContest dataset” to indicate its relevance more explicitly.
>
> ### Reproducibility
> We will make all code and necessary parameters available to the public (currently they can be accessed in footnote 1.). We believe that our work is highly reproducible (and some other groups have already used it for evaluation)!

---

### Meta-Review · Area_Chair_qv3G · 2023-09-15

**Recommendation:** 4

**Metareview:**

The paper introduces ODEX, a pioneering Open-Domain EXecution-based dataset designed for natural language to Python code generation. ODEX features 945 NL-Code pairs that span 79 diverse libraries and are supported in four languages: English, Spanish, Japanese, and Russian. These pairs are sourced from StackOverflow, ensuring relevance to real-world coding queries. Additionally, the dataset includes 1,707 human-written test cases to facilitate execution-based evaluation. A notable observation from the study is the differing behaviors of leading code language models. While CODEX outperforms in general, CODEGEN shows improvement with scaling. The paper highlights the performance gaps of these models in open and closed domains. Reviewers commend ODEX for its "open-domain" nature, execution-based evaluation support, and multilingual capability. They recognize its potential to address challenges in open-domain code execution and appreciate its broader coverage of libraries compared to existing solutions. The dataset is expected to significantly benefit the code generation research community.

The paper presents a significant contribution to the community by introducing a dataset with human-written test cases, emphasizing execution-based evaluation, which aligns more closely with human preferences than n-gram based evaluations. The meticulous process of dataset creation, including a thorough quality check during annotation, is commendable. The dataset's strength lies in its "open-domain" nature, derived from genuine and practical queries, necessitating diverse library imports and usage. This diversity is an important feature, setting ODEX apart from other datasets. The paper provides a detailed account of the dataset collection and annotation procedure, offering innovative solutions to challenges like irreproducible runs, randomized outputs, and specialized equivalence checks. Benchmark experiments offer valuable insights, especially the intriguing result comparing the performance gap between Codex and Codegen in open versus closed domains as model size grows. Overall, the dataset's potential for superior data quality compared to existing datasets makes it a worthy addition to the field.

The authors must consider the following weaknesses: a) The process of showing both the query and canonical code to annotators when writing test cases could introduce bias, potentially limiting the test cases to the canonical code solution. b) The paper's experiments do not conclusively demonstrate the sufficiency of the test cases. c) The dataset's small size, especially when divided by language, raises concerns about the robustness of the results.

---

### Decision · Program_Chairs · 2023-10-07

**Decision:**

Accept-Findings

**Comment:**

The paper introduces ODEX, a pioneering Open-Domain EXecution-based dataset designed for natural language to Python code generation. ODEX features 945 NL-Code pairs that span 79 diverse libraries and are supported in four languages: English, Spanish, Japanese, and Russian. These pairs are sourced from StackOverflow, ensuring relevance to real-world coding queries. Additionally, the dataset includes 1,707 human-written test cases to facilitate execution-based evaluation. A notable observation from the study is the differing behaviors of leading code language models. While CODEX outperforms in general, CODEGEN shows improvement with scaling. The paper highlights the performance gaps of these models in open and closed domains. Reviewers commend ODEX for its "open-domain" nature, execution-based evaluation support, and multilingual capability. They recognize its potential to address challenges in open-domain code execution and appreciate its broader coverage of libraries compared to existing solutions. The dataset is expected to significantly benefit the code generation research community.

The paper presents a significant contribution to the community by introducing a dataset with human-written test cases, emphasizing execution-based evaluation, which aligns more closely with human preferences than n-gram based evaluations. The meticulous process of dataset creation, including a thorough quality check during annotation, is commendable. The dataset's strength lies in its "open-domain" nature, derived from genuine and practical queries, necessitating diverse library imports and usage. This diversity is an important feature, setting ODEX apart from other datasets. The paper provides a detailed account of the dataset collection and annotation procedure, offering innovative solutions to challenges like irreproducible runs, randomized outputs, and specialized equivalence checks. Benchmark experiments offer valuable insights, especially the intriguing result comparing the performance gap between Codex and Codegen in open versus closed domains as model size grows. Overall, the dataset's potential for superior data quality compared to existing datasets makes it a worthy addition to the field.

The authors must consider the following weaknesses: a) The process of showing both the query and canonical code to annotators when writing test cases could introduce bias, potentially limiting the test cases to the canonical code solution. b) The paper's experiments do not conclusively demonstrate the sufficiency of the test cases. c) The dataset's small size, especially when divided by language, raises concerns about the robustness of the results.